# MarDini: Masked Auto-regressive Diffusion for Video Generation at Scale

**Haozhe Liu**[1,2,*],    **Shikun Liu**[2,†],    **Zijian Zhou**[2,†],    **Mengmeng Xu**[2],    **Yanping Xie** [2],    **Xiao Han**[2],    **Juan C. Pérez**[2],    **Ding Liu**[2],    **Kumara Kahatapitiya**[2],    **Menglin Jia**[2],    **Jui-Chieh Wu**[2],    **Sen He**[2],    **Tao Xiang**[2],    **Jürgen Schmidhuber**[1],    **Juan-Manuel Perez-Rua**[2,*]

[1] *KAUST*

[2] *Meta AI*

[†] *Equal Contribution*

[*] *Correspondence: haozhe.liu@kaust.edu.sa; jmpr@meta.com*

**Reviewed on OpenReview:** *https://openreview.net/forum?id=fuOHI59rUW*

## Abstract

We introduce MarDini, a new family of video diffusion models that integrate the advantages of masked auto-regression (MAR) into a unified diffusion model (DM) framework. Here, MAR handles temporal planning, while DM focuses on spatial generation in an asymmetric network design: i) a MAR-based planning model containing most of the parameters generates planning signals for each masked frame using low-resolution input; ii) a lightweight generation model uses these signals to produce high-resolution frames via diffusion de-noising. MarDini's MAR enables video generation conditioned on any number of masked frames at any frame positions: a single model can handle video interpolation (e.g., masking middle frames), image-to-video generation (e.g., masking from the second frame onward), and video expansion (e.g., masking half the frames). The efficient design allocates most of the computational resources to the low-resolution planning model, making computationally expensive but important spatio-temporal attention feasible at scale. MarDini sets a new state-of-the-art for video interpolation; meanwhile, within few inference steps, it efficiently generates videos on par with those of much more expensive advanced image-to-video models.

## 1 Introduction

Auto-regressive (AR) models (Vaswani et al., 2017; Peng et al., 2023; Schmidhuber, 1992b; Schlag et al., 2021) have recently demonstrated remarkable success in natural language processing (Dubey et al., 2024; Team et al., 2023; Achiam et al., 2023), sparking efforts to achieve similar breakthroughs in computer vision (Sun et al., 2024; He et al., 2021; Deng et al., 2024; Shi et al., 2024; Wang et al., 2024d; Tian et al., 2024). However, unlike the *discrete, sequential, and easily tokenized* nature of language, visual data consist of *continuous* pixel signals distributed across a *high-dimensional* space, making them more difficult to model through 1D auto-regression.

To overcome this challenge, recent studies have explored vector quantization techniques (Van Den Oord et al., 2017; Razavi et al., 2019) to convert continuous pixel data into discrete representations suitable for AR modelling. Unfortunately, these approaches (Yu et al., 2022; Ramesh et al., 2021) rely on *causal attention*, which is not well aligned for high-dimensional visual data, often leading to diminished performance (Li et al., 2024), particularly on large-scale datasets (Xie et al., 2024; Zhou et al., 2024). To mitigate this limitation, masked auto-regression (MAR) has been introduced (Chang et al., 2022; Li et al., 2023a). MAR replaces the causal attention with *bi-directional attention* (He et al., 2021; Devlin et al., 2019), effectively simulating auto-regressive behaviour while being more capable of handling visual data. Leveraging this approach, MAR exhibits flexibility in handling diverse generation tasks through different masking strategies, such as image generation (Chang et al., 2022; Li et al., 2023a), out-painting (Chang et al., 2022), video expansion (Yu

et al., 2023a) and class-conditioned video generation (Yu et al., 2024; Voleti et al., 2022) while maintaining manageable computational overhead. Although MAR shows potential in scaling image and video generation tasks (Chang et al., 2023; Yu et al., 2023a; 2024), its key bottleneck lies in its training instability and weak performance which is tied to the reliance on discrete representations (Ramesh et al., 2021; Razavi et al., 2019; Li et al., 2024).

Meanwhile, Diffusion models (DMs) (Ho et al., 2020; Neal, 2001; Jarzynski, 1997) have emerged as a successful alternative for scaling vision generative models, offering stable training by modelling visual signals directly in a continuous space. However, DMs tend to incur high inference costs due to the requirement of the multi-step diffusion process. Here, video generation poses an even greater challenge — Video is a *strict super-set* of the image domain, requiring additional modelling for temporal consistency and complex motion dynamics.

To this end, we propose a new paradigm for video generation that combines the flexibility of MAR in a *continuous* space with the *robust generative capabilities* of DM. Specifically, we present a scalable training recipe and an efficient neural architecture design for video generation. Our model decomposes video generation into two sub-tasks — temporal and spatial modelling — handled by *distinct* networks with *an asymmetric design* based on the following two principles:

1. *MAR handles long-range temporal modelling, while DM focuses on detailed spatial modelling.*

2. *MAR operates with more parameters at a lower resolution, while DM operates with fewer parameters at a higher resolution.*

Following these principles, we use the same training batch for both MAR and DM but employ two distinct processes operating at different resolutions. MAR receives randomly masked low-resolution input frames and predicts the corresponding planning signals. Conditioned on these planning signals via cross-attention and the unmasked frames, DM learns to incrementally recover the masked high-resolution frames from noise. Finally, we introduce a progressive training strategy that gradually curates mask ratios and with its data pipelines, allowing our model to be *trained from scratch on unlabeled video data.* This eliminates the common reliance on text-to-image and text-to-video pre-training, as seen in other video diffusion models (Girdhar et al., 2023; Blattmann et al., 2023a).

Our model integrates MAR-based planning signals with a DiT-based (Peebles & Xie, 2023; Chen et al., 2024c) lightweight, tiny diffusion model, hence the name **MarDini**. Our empirical study on MarDini highlights the following key characteristics:

- **Flexibility.** With MAR conditioning, MarDini naturally supports a range of video generation tasks through flexible masking strategies. For example, when given the first frame and masking the rest, it performs image-to-video generation; when given a video and masking subsequent frames, it performs video expansion; and, when given the first and last frames and masking the middle frames, it performs video interpolation. By hierarchically and auto-regressively masking middle frames across multiple inferences, MarDini generates slow-motion videos.

- **Scalability.** MarDini can be trained from scratch at scale, without relying on generative image-based pre-training. This approach enables the model to scale from video interpolation to full video generation, directly bypassing the need for image-based pre-training. Such observation may accelerate the prototype verification for new video generation models.

- **Efficiency.** MarDini's asymmetric design allocates more computational resources to lower resolutions, making it memory-efficient and fast during inference. With lower overall memory usage, MarDini allows the deployment of computationally intensive spatio-temporal attention mechanisms at scale, improving its ability to model complex motion dynamics.

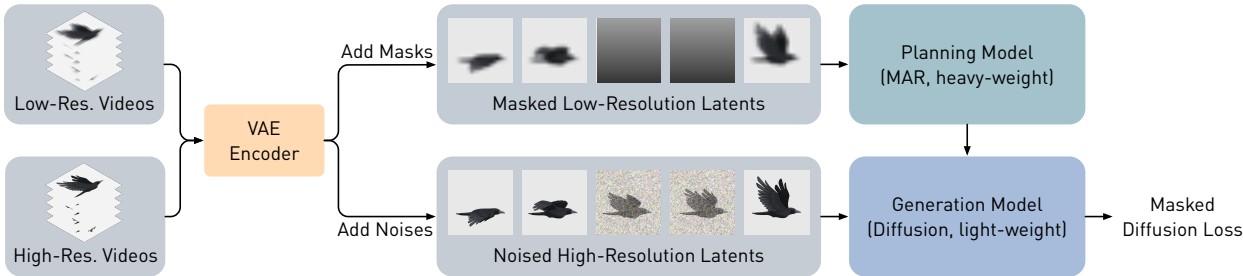

Figure 1: **MarDini Training Pipeline Overview.** A latent representation is computed for unmasked frames that serve as a conditional signal to a generative process. On the first hand, we have a planning model that autoregressively encodes global conditioning signals from a low-resolution version of the unmasked latent inputs. On the other hand, the planning signals are fed to the diffusion-based generation model through cross-attention layers. A high-resolution version of the input conditions is also ingested by the diffusion model, enabling generation with a coherent temporal structure and a direct mechanism to attend to fine-grained details of the unmasked frames. MarDini is trained end-to-end via masked frame-level diffusion loss. Note that the latent code extracted by the VAE is highly compressed; we use the raw frame as an approximate proxy for clearer visualization.

## 2 MarDini: An Efficient and Asymmetric Video Diffusion Model

### 2.1 Design Overview

MarDini is a video generation model designed to efficiently generate high-resolution videos using an asymmetric network architecture. As shown in Figure 1, MarDini consists of two networks: a heavy-weight MAR *planning* model and a light-weight *generation* DM. During training, the planning network processes randomly masked low-resolution frames and predicts corresponding planning signals. These planning signals compress the semantic and long-range temporal information, guiding the DM's high-resolution generation process. The DM receives noisy frames at the masked positions and reconstructs them by progressively removing noise.

In this section, we outline and address the key design challenges involved in training MarDini. First, we describe the data representations and their corresponding notations within the MarDini framework (Section 2.2). Next, we describe the design details of the MAR planning network and the DM, along with the integration of additional guidance such as diffusion steps and planning signals (Section 2.3). Finally, we outline the multi-stage training recipe for MarDini, which we found to be essential for ensuring stable training (Section 2.4). Collectively, these innovations enable MarDini to become one of the first video generation models capable of being trained from scratch using only unlabelled video data.

### 2.2 Data Representation and Notations

**VAE Compressor.** Consistent with prior works (Dai et al., 2023a; Girdhar et al., 2023), we adopt a pre-trained Variational Auto-Encoder (VAE) (Kingma & Welling, 2014), denoted by $\mathcal{D}_{\text{enc}}$, to compress videos into a low-dimensional continuous latent space, which improves both training and inference efficiency. Our VAE employs a 16-channel latent dimension with an $8\times$ spatial compression rate to preserve spatial details, following Dai et al. (2023a). The VAE outputs are then patchified into a shape of $N \times C$, where $N$ represents the token count and $C = 16$ represents its latent dimension.

**MAR Planning Model.** Given a low-resolution input video $\mathbf{X}_{\text{low}} = \{x_i^{\text{low}}\}_{i=1:K}$ with $K$ frames, we apply the VAE encoder to compress the frames into their corresponding latent representations: $\mathbf{Z}_{\text{low}} = \{z_i^{\text{low}}\}_{i=1:K} = \mathcal{D}_{\text{enc}}(\mathbf{X}_{\text{low}})$. To train the MAR planning model $\mathcal{P}$, we randomly select $K' < K$ video latents $\{z_j^{low}\}_{j=1:K'} \in \mathbf{Z}_{\text{low}}$ and replace them with a learnable mask token [MASK], resulting in the final masked low-resolution latent inputs $\mathbf{Z}_{\text{low}}^{\text{mask}}$. The planning model then processes $\mathbf{Z}_{\text{low}}^{\text{mask}}$ and predicts $\mathbf{Z}_{\text{cond}} = \mathcal{P}(\mathbf{Z}_{\text{low}}^{\text{mask}}) = \{z_i^{\text{cond}}\}_{i=1:K}$, where $z_i^{\text{cond}}$ is the planning signal for the $i$-th frame, shaped as $N_{\text{low}} \times C_{\text{low}}$, with $N_{\text{low}}$ representing the number of patches per frame.

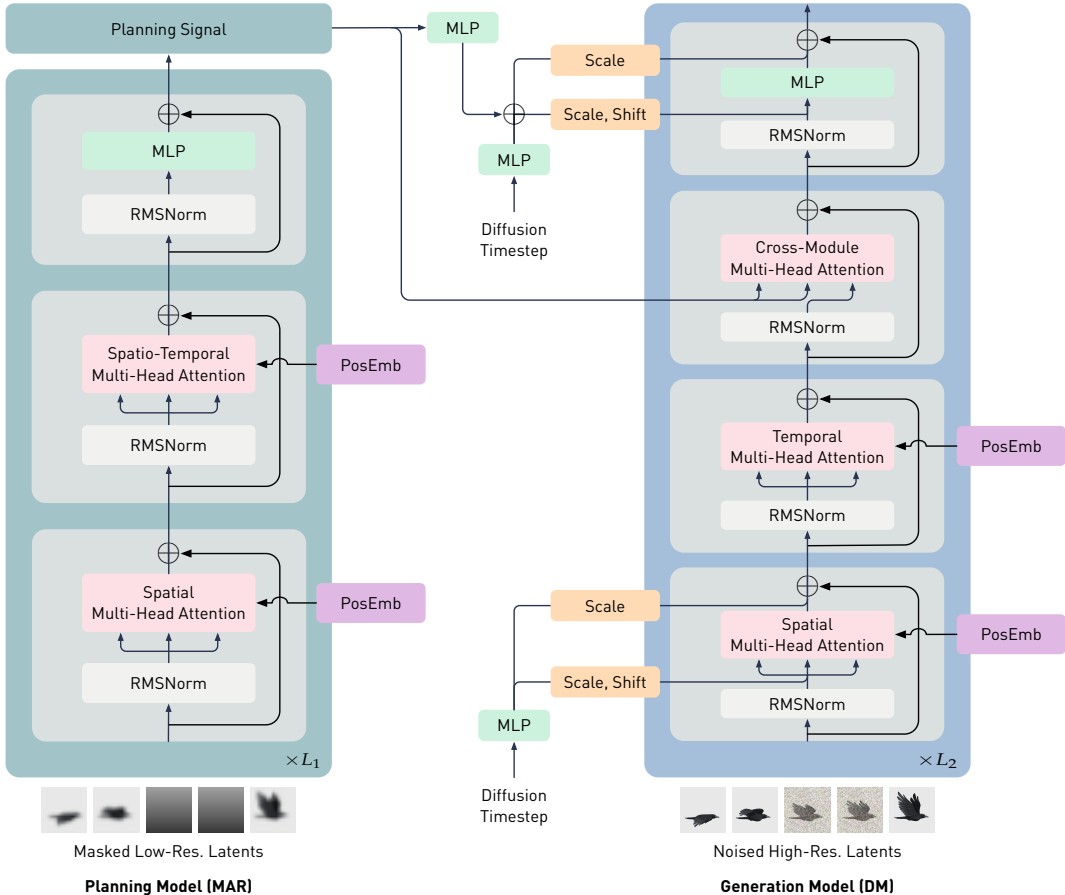

Figure 2: **MarDini Design Details.** MarDini employs a transformer architecture for both the planning and generation models, incorporating a DiT-style block for the generation model and a Llama-style block for the planning model. We set $L_1 \gg L_2$, where $L_1$ and $L_2$ refer to the number of layers in the planning and generation model respectively. The "Diffusion Timestep" below matches the one above, and we draw them separately to avoid line crossings and interference for clarity.

**DM Generation Model.** Conversely, we obtain high-resolution video latents $\mathbf{Z}_{\text{high}} = \mathcal{D}_{\text{enc}}(\mathbf{X}_{\text{high}})$ with dimensions $N_{\text{high}} \times C_{\text{high}}$, generated by the VAE encoder using the same video inputs at high resolution: $\mathbf{X}_{\text{high}} = \{x_i^{\text{high}}\}_{i=1:K}$. Notably, we have $N_{\text{high}} \gg N_{\text{low}}$. At diffusion step $t$, we sample noise and add it to $K'$ frames that were masked in the planning model (denoted by [NOISE]), leaving the remaining $K - K'$ reference frames unchanged (denoted by [REF]). This produces the final noisy high-resolution video latent inputs $\mathbf{Z}_{\text{high}}^{\text{noise},t}$. Then, the generation model $\mathcal{G}$ processes these latent inputs $\mathbf{Z}_{\text{high}}^{\text{noise},t}$ and performs a standard denoising step, where we denote the DM output at time step $t$ as $\mathcal{G}(\mathbf{Z}_{\text{high}}^{\text{noise},t}, \mathbf{Z}_{\text{cond}}, t)$.

## 2.3 Architecture Design

In this section, we provide a comprehensive explanation of the MarDini architecture, including its detailed design, model configurations, and variations.

### 2.3.1 MarDini Block Design

Figure 2 illustrates the design of the MarDini's MAR and DM models, both of which are based on the transformer architecture (Vaswani et al., 2017).

In the MAR planning model, we adhere to the design conventions established in Llama models (Dubey et al., 2024; Touvron et al., 2023), which apply RMS-Norm (Zhang & Sennrich, 2019) to normalize the

inputs of each attention block. Additionally, layer normalization (Ba et al., 2016) is applied to normalize the projected features in multi-head attention, further enhancing training stability. Due to the use of low-resolution inputs, we manage to directly employ spatio-temporal attention, allowing tokens to attend across frames. This design is feasible only with asymmetric resolution inputs, as it prevents excessive memory consumption during training.

Concretely, within each attention block in MAR, we utilize rotary positional encoding (RoPE) (Su et al., 2024) to encode both the spatial and temporal positions of the video tokens. To accomplish this, we apply a 2D RoPE to encode the 3-dimensional video data. Specifically, we flatten the image patches into a 1-dimensional token sequence and insert a learnable `[NEXT]` token to differentiate image patches across different rows, akin to the [SEP] in Devlin et al. (2019) and [nextline] in Gao et al. (2024). This design effectively handles video data with varying aspect ratios and resolutions.

We design the DM model in alignment with MAR, but with three key differences. First, we adopt a DiT-style approach (Peebles & Xie, 2023), using AdaIN (Huang & Belongie, 2017) to incorporate the diffusion steps as a conditional signal. Second, we introduce a cross-attention layer to process the planning features predicted by the MAR model. Lastly, we replace spatio-temporal attention with temporal attention (Blattmann et al., 2023b) to reduce the computational cost associated with high-resolution inputs in DM.

### 2.3.2 Identity Attention

In our initial experiments, we observed significant training instability in MarDini's DM. We speculate that this is due to two main factors: i) the inherent distributional disparity between noisy (`[NOISE]`) tokens and clean reference (`[REF]`) tokens, which is further amplified by the stochastic nature of sampling diffusion steps; and ii) the random positions and varying lengths of these `[NOISE]` tokens. These factors likely compound, potentially disrupting the DM's training signals and hindering the model's ability to converge efficiently.

To address this challenge, we introduce Identity Attention, which enables the model to easily distinguish between `[REF]` and `[NOISE]` tokens by employing a separate attention strategy. As illustrated in Figure 3, `[REF]` tokens simply serve as an *identity projection*, preserving the input reference frames without attending to other tokens. In contrast, `[NOISE]` tokens possess a global view, attending to tokens across *all frames*. The `[REF]` tokens serve as guidance for generation, so we design them to be isolated from other tokens, while `[NOISE]` tokens provide global attention to all conditional signals for generation. We incorporate Identity Attention in both the spatio-temporal layers of MAR and the temporal layers of DM, which has been found to significantly enhance training stability in both models.

Based on the above design, we present four model variants with distinct configurations, detailed in Appendix A.

Figure 3: **Identity Attention Design Details in DM.** In this setup, `[REF]` tokens only attend to themselves, while `[NOISE]` tokens attend to all other tokens.

### 2.4 MarDini Training Recipes

In this section, we outline the training pipeline of MarDini. Specifically, we employ a multi-stage progressive training strategy that gradually increases task difficulty. This approach offers two key benefits: i) progressive learning inherently enhances training stability and improves the performance of generative models, as demonstrated by Karras (2018) and Chen et al. (2024b); and ii) it allows for the collection of checkpoints from earlier stages, which helps mitigate setbacks caused by suboptimal configurations. Below, we elaborate on our detailed progressive training strategy, including the training objectives, architecture design, and training data configurations. A comprehensive training manual for MarDini can be found in the Appendix B.

### 2.4.1 Training Tasks: From Frame Interpolation to Video Generation

Our training objectives are organized into three stages: i) Initial Stage: We separately train the planning and generation models, each with its own learning objective, to initialize their model weights. ii) Joint-Model Stage: We combine the models for joint training on a simple video interpolation task, using only a masked diffusion loss. iii) Joint-Task Stage: We further train the model by gradually reducing the number of preserved reference frames, enabling it to jointly learn video interpolation and image-to-video generation tasks.

**Initial Stage.** Wang et al. (2024a) pointed out that transformers with a large parameter count often experience unstable training. As such, we simplify the training dynamics by separately warming up the two models as an initial step.

To optimize generation model $\mathcal{G}$, we employ a masked diffusion loss $\mathcal{L}_{\mathrm{DM}}$:

$$\mathcal{L}_{\mathrm{DM}}^{\theta} = ||\mathbf{M} \cdot \mathbf{V}^t - \mathbf{M} \cdot \mathcal{G}_\theta(\mathbf{Z}_{\mathrm{high}}^{\mathrm{noise},t}, \mathbf{Z}_{\mathrm{uncond}}, t)||_2^2, \tag{1}$$

where $\mathbf{Z}_{\mathrm{uncond}}$ is a learnable token serving as *unconditional guidance* from the planning model. $\theta$ represents the parameters of the generation model, and $\mathbf{M}$ denotes the binary masks used to mask out all clean reference frames. Inspired by Blattmann et al. (2023b); Salimans & Ho (2022), we apply velocity prediction as the diffusion loss, where the prediction target $\mathbf{V}^t = \{v_i^t\}_{i=1:K}$ represents the velocity at time step $t$ for the $i$-th frame, defined as $v_i^t = \alpha_t \epsilon - \sigma_t z_i^h, \epsilon \sim \mathcal{N}(0, I)$. Here, $\alpha_t$ and $\sigma_t$ correspond to the diffusion scheduler at $t$ step.

To optimize MAR planning model $\mathcal{P}$, we employ a masked reconstruction loss $\mathcal{L}_{\mathrm{MAR}}$:

$$\mathcal{L}_{\mathrm{MAR}}^{\phi,\zeta} = ||\mathbf{M} \cdot \mathbf{Z}_{\mathrm{low}} - \mathbf{M} \cdot f_\zeta(\mathcal{P}_\phi(\mathbf{Z}_{\mathrm{low}}^{\mathrm{mask}})||_2^2. \tag{2}$$

where $f$ denotes a projection layer that depatchifies the model predictions to match the resolution of the low-resolution input image $\mathbf{Z}_{\mathrm{low}}$. $\phi, \zeta$ represent the learnable parameters of the planning model and the projection layer respectively. Note that, $f$ is only used during the initial training stage, and will be removed in the later training stages.

**Joint-Model Stage.** After the initial pre-training stage, we then jointly train the planning and generation models end-to-end using a unified masked diffusion learning objective $\mathcal{L}_{\mathrm{MDiff}}$:

$$\mathcal{L}_{\mathrm{MDiff}}^{\theta,\phi} = ||\mathbf{M} \cdot \mathbf{V}^t - \mathbf{M} \cdot \mathcal{G}_\theta(\mathbf{Z}_{\mathrm{high}}^{\mathrm{noise},t}, \mathcal{P}_\phi(\mathbf{Z}_{\mathrm{low}}^{\mathrm{mask}}), t)||_2^2, \tag{3}$$

where $\mathbf{Z}_{\mathrm{cond}} = \mathcal{P}(\mathbf{Z}_{\mathrm{low}}^{\mathrm{mask}})$ is the planning signal predicted by MAR. In order to enable classifier-free guidance (Ho & Salimans, 2022) on the planning signal, we maintain a fixed probability of $^1/_{10}$ to randomly replace $\mathbf{Z}_{\mathrm{cond}}^t$ with $\mathbf{Z}_{\mathrm{uncond}}$.

**Joint-Task Stage.** In the final training stage, we reuse the learning objective from the previous stage, but gradually increase the masking ratio to induce more challenging generation tasks. This stage requires a significantly larger computational resources with higher-resolution videos, as it determines the model's final performance. By gradually increasing the masking ratios, we smoothly transform the model's task from video interpolation to single-image-to-video generation. This procedure ultimately enables the model to generate videos with a variable number of input frames at arbitrary temporal locations.

### 2.4.2 DM Architecture: From Spatio-Temporal to Temporal Attention

In conjunction with our progressive training objectives, we also introduce a progressive architectural design. Specifically, we first use spatio-temporal attention in the DM during the initial training stage. This choice promotes convergence, compared to temporal attention, as noted in Gao et al. (2024). Since in our initial stage we train the DM in isolation and on a relatively low-resolution setup, this sophisticated attention incurs in minor computational overhead. When integrating MAR with the DM in the second stage, we replace the spatio-temporal attention with the more cost-effective temporal attention, thus increasing the efficiency of the generation model.

### 2.4.3   Data: Progressive Configuration of Specifications

Analogous to our progressive strategies for training objective and architecture we also propose a progressive data configuration. Over time, we gradually increase the video's spatial resolution, alongside progressively extending the video's duration. This approach ensures efficient use of computational resources and facilitates effective model scaling, allowing MarDini to handle more complex and high-resolution video data as training progresses.

The details of the mask ratios, architecture design and data size used within each training stage are reported in Appendix B.

## 3   Experiments

We evaluate MarDini on two benchmarks: VIDIM-Bench (Jain et al., 2024), for long-term video interpolation, and VBench (Huang et al., 2024) for image-to-video generation. We further elaborate on the specifics of these benchmarks in Appendix D. *We highly encourage referring to the generated videos in the supplementary material for a comprehensive understanding of the quality of the generated videos.*

### 3.1   Ablation Studies and Analysis

**Effectiveness of MAR and DM.**   We first demonstrate the importance of having a DM on top of our MAR planning model. In fact, it is tempting to hypothesize that MAR on its own contains all the ingredients to enable high-quality video interpolation. To explore this, we introduce a projection layer to directly unpatchify the output of the MAR model without intermediate diffusion with DM. Our experiments on VIDIM-Bench reveal that, MAR on its own, performs poorly on interpolation tasks, as shown by the first two and last two rows in Table 1, for both the 1B and 3B settings. This result suggests that directly applying MAR to continuous space is suboptimal, a result consistent with previous findings (Li et al., 2024). Similarly, directly tackling this task with a small DM without global guidance, according to the third row of Table 1, results in sub-optimal performance. However, by combining MAR's planning capability with DM's stable performance in continuous space, we achieve optimal results, demonstrating that both components are beneficial for video generation.

Table 1: **Effectiveness of MAR and DM design.** The reported results are FVD on DAVIS-7 and UCF101-7 from VIDIM-Bench. All experiments are evaluated at a resolution of $[256 \times 256]$ using DDIM scheduler with 25 steps.

| Planning Model | Generation Model | FVD↓ | |
|---|---|---|---|
| | | DAVIS | UCF101 |
| MAR-1B | - | 427.66 | 741.80 |
| MAR-3B | - | 373.03 | 701.03 |
| - | DM-0.3B | 320.89 | 383.04 |
| MAR-1B | DM-0.3B | 224.07 | 258.08 |
| MAR-3B | DM-0.3B | **102.87** | **197.69** |

Table 2: **Efficiency of the MarDini's generations with and without the asymmetric design.** Both latency and GPU memory is measured as the average time to generate a video using DDIM with 25 steps using a single A100 GPU, and with `bf16` mixed precision.

| Asymm. Attention | Asymm. Resolution | # Inference Frames | [256 × 256] | | [512 × 512] | |
|---|---|---|---|---|---|---|
| | | | Latency | GPU Mem. | Latency | GPU Mem. |
| ✗ | ✗ | 9 (1 to 8) | 2.76 s | 25.22 G | 25.09 s | 74.44 G |
| ✗ | ✓ | 9 (1 to 8) | | | 17.91 s | 41.03 G |
| ✗ | ✗ | 13 (1 to 12) | 4.41 s | 27.80 G | Out of Memory | |
| ✗ | ✓ | 13 (1 to 12) | | | 34.58 s | 62.51 G |
| ✓ | ✗ | 13 (1 to 12) | **2.63 s** | **27.75 G** | Out of Memory | |
| ✓ | ✓ | 13 (1 to 12) | | | **6.05 s** | **42.57 G** |

**Efficiency Analysis.**   Table 2 illustrates latency and memory usage across different input resolutions and frame lengths, measured on the same computational platform. When MAR is set to operate symmetrically with the DM with the same inputs, the model cannot fit in the available GPU memory as we increase the resolution and/or number of frames. In contrast, our asymmetric design enables the generation of 12-frame clips at 512 resolution in just a few seconds. The rapid generation process is partially attributed to the DM requiring relatively few inference steps to converge, thanks to the well-structured planning signal it receives, as shown in Figure 5a. Notably, inference speed could be further optimized, as the only acceleration technique we incorporated during our experiments is mixed precision, without employing caching strategies (Liu et al., 2024; Zhao et al., 2024), FSDP-based parameter/data parallelization (Chen et al., 2024d), or

static compilation of the underlying computational graph. Similarly, memory usage could be further reduced through CPU offloading, sliced attention, sequential VAE inference, *etc.*

**Explaining MAR's Planning Signal.** We provide an intuitive explanation of MAR's role in MarDini. During training, a learnable token is used to randomly replace MAR to support CFG (Ho & Salimans, 2022), allowing DM to generate videos independently. We visualize the results of MarDini with and without MAR's planning signals. As shown in Figure 4, without the planning model, DM can still produce meaningful frames but, as expected, lacks "global planning." For example, in Figure 4 (Left), DM moves objects in different directions, causing distortion in the building, which suggests a weaker or non-existing prior model of how objects move. Similarly, in Figure 4 (Right), DM fails to accurately predict the movement of the fire. In contrast, incorporating the planning signal addresses these visual flaws. These results indicate that MAR's planning signal effectively hints how elements should move, ensuring long-term coherence in the generated video.

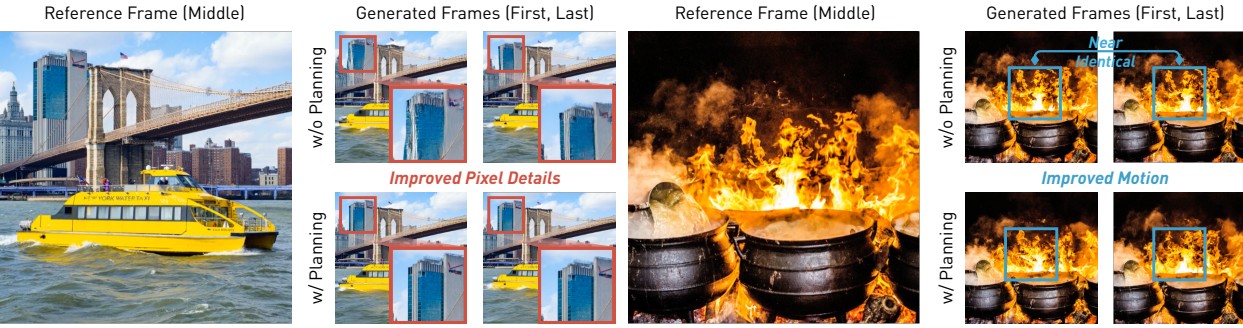

Figure 4: **MarDini's generations with and without the planning model.** Here we show video frames generated when conditioning on the middle frame. Without MAR's planning signal, DM generates degraded motion, such as pixel distortions (highlighted in **red**, left) or incorrect motions (highlighted in **blue**, right).

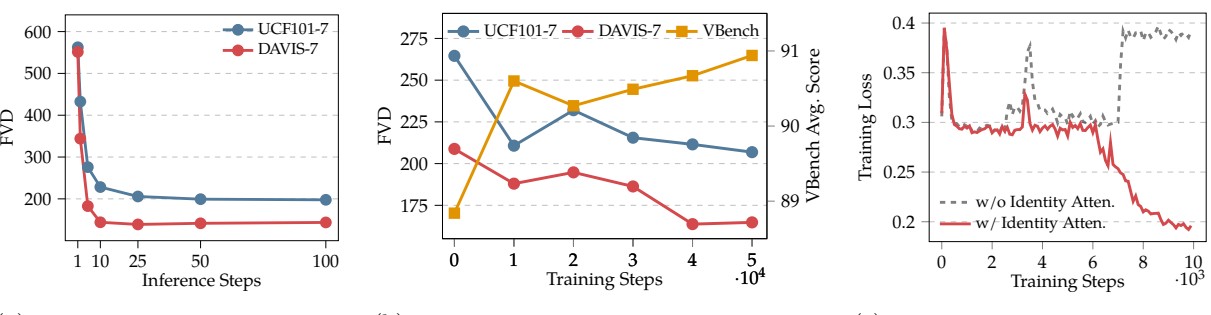

(a) Video interpolation results with varying inference steps.

(b) Relationship between video interpolation and image-to-video generation.

(c) Training loss of MarDini w and w/o Identity Attention.

Figure 5: **MarDini Training and Inference Performance.** (a) MarDini-S/ST-512 achieves optimal generation performance with few inference steps using the DDIM solver; (b) As training progresses, MarDini shows improvement in the tasks of both video interpolation and image-to-video. These results are based on a mask ratio ranging from 0.15 to 0.6 for 9-frame generation; and (c) The design of Identity Attention is crucial for stable training convergence in MarDini during the initial training stage; without it, the model fails to converge.

**From Video Interpolation to Image-To-Video Generation.** Our training recipe follows the philosophy of transitioning from video interpolation to image animation. Herein, we empirically demonstrate that these two tasks are related, validating the soundness of our pipeline. As shown in Figure 5b, we track the performance of MarDini on both video interpolation and image animation during a training phase aimed at scaling the resolution from 256 to 512. This stage marks the first point during training where the model successfully performs both tasks simultaneously. We observe a promising consistency between the performance of image animation and video interpolation, providing solid evidence that these tasks do not hinder each other. Furthermore, with a carefully tuned mask ratio, the model can be trained in a unified manner to efficiently achieve both tasks.

**Impact of Identity Attention.** We explore the effectiveness of Identity Attention in handling our specific data format, which integrates both reference frames and noised frames into a single sequence. As illustrated in Figure 5c, we track the training trajectory in the early stages of the DM generation model. We recognize that this type of input can lead to unstable training, particularly when starting from scratch, as the differences between reference frames are difficult to discern. However, the proposed Identity Attention mechanism mitigates this instability. The decrease in training loss observed after 6K steps is attributed to the use of a warm-up learning rate, where the learning rate is intentionally kept low during the initial steps.

## 3.2 Results on Video Interpolation

Table 3: **Performance of zero-shot video interpolation on VIDIM-Bench.** The reported results are taken directly from VIDIM (Jain et al., 2024). AMT, RIFE, and FILM are single-inference methods, while LDMVFI, VIDIM, and our approach are based on diffusion models with multiple inference steps. MidF-SSIM and MidF-LPIPS represent the SSIM and LPIPS scores, respectively, for the middle frame. For MarDini-512, we downscale the generated videos to 256 resolution for a fair comparison.

| Method | DAVIS-7 | | | | UCF101-7 | | | |
|---|---|---|---|---|---|---|---|---|
| | MidF-SSIM | MidF-LPIPS | FID | FVD | MidF-SSIM | MidF-LPIPS | FID | FVD |
| AMT (Li et al., 2023b) | 0.4853 | 0.2865 | 34.65 | 234.50 | 0.7903 | 0.1691 | 31.60 | 344.50 |
| RIFE (Huang et al., 2022) | 0.4546 | 0.2954 | 23.98 | 240.04 | 0.7769 | 0.1564 | 18.72 | 323.80 |
| FILM (Reda et al., 2022) | 0.4718 | 0.3048 | 30.16 | 214.80 | 0.7869 | 0.1620 | 26.06 | 328.20 |
| LDMVFI (Danier et al., 2024) | 0.4175 | 0.2765 | 22.10 | 245.02 | 0.7712 | **0.1564** | **18.09** | 316.30 |
| VIDIM (Jain et al., 2024) | 0.4221 | 0.2986 | 28.06 | 199.32 | 0.6880 | 0.1768 | 34.48 | 278.00 |
| MarDini-S/ST-256 | 0.4249 | 0.3654 | 49.21 | 224.07 | 0.7654 | 0.2480 | 45.85 | 258.08 |
| MarDini-L/ST-256 | 0.4959 | 0.2768 | **20.64** | 102.87 | 0.7734 | 0.2213 | 28.85 | **197.69** |
| MarDini-S/ST-512 | 0.5017 | 0.3193 | 25.92 | 138.86 | **0.7960** | 0.2315 | 30.24 | 205.71 |
| MarDini-L/ST-512 | **0.5314** | **0.2736** | 20.76 | **99.05** | 0.7814 | 0.2347 | 30.08 | 204.20 |
| MarDini-L/T-512 | 0.5085 | 0.3083 | 25.30 | 117.13 | 0.7893 | 0.2270 | 30.72 | 198.94 |

We compare MarDini with the existing methods on the VIDIM benchmark (Jain et al., 2024) for video interpolation, where the task is to generate 7 frames between a starting and an ending conditional frames. As shown in Table 3, MarDini achieves competitive performance among different evaluation metrics. Since FVD is the only metric that accounts for the temporal modelling, we prioritize it in our evaluation. Notably, MarDini outperforms other baselines in this metric, achieving state-of-the-art performance. Notably, MarDini-L/T employs an asymmetric attention mechanism, where the planning model utilizes spatio-temporal attention, while the generation model relies on temporal attention. Compared to the model that uses spatio-temporal attention for both models (MarDini-L/ST), the results suggest that the asymmetric attention mechanism does not significantly affect performance, achieving a satisfactory trade-off between efficiency and quality. We provide additional visualizations in Appendix C and the supplementary materials.

## 3.3 Results on Image-to-Video Generation

In this section, we evaluate our model's single-image-to-video generation capabilities in comparison with other methods using the VBench dataset (Huang et al., 2024). As shown in Table 4, our method performs competitively, especially in terms of latency, despite incorporating expensive spatio-temporal attention. For fairness, latency is calculated with the same resolution. In this study, we focus on validating the soundness of our proposed roadmap, only considering the initial pre-training stage rather than delving into post-training techniques. As a result, we do not incorporate additional conditional signals such as language instructions or motion score guidance. Therefore, direct comparisons on video quality, particularly in relation to dynamic degree, are not entirely fair. However, we fully report these numbers for reference.

We also report the results on the benchmark without the motion score (referred to as Dynamic Degree in VBench). All evaluation metrics are detailed in Appendix D. The empirical study shows MarDini's strong potential, performing on par with other existing methods across several metrics while exhibiting higher efficiency and requiring no generative image pre-training. Among these baselines, SVD-XL can be regarded as a DM-only alternative, with a parameter size comparable to MarDini-S. Notably, while MarDini-S achieves comparable performance to SVD-XL, it offers greater flexibility and efficiency without compromising generation quality. Interestingly, we observe that MarDini-S marginally outperforms MarDini-L on some

Table 4: **Image-to-Video Performance on VBench.** The reported results of baseline methods are sourced from VBench (Huang et al., 2024). For fair latency comparison, we standardize the input size to [512×512] for low and medium resolutions, and [768×768] for high resolution cases across all methods. All other metrics were collected using the original resolutions reported in the first column.

| Method | Frame Resolution | Image-based Pre-training | Latency (s/frame) | I2V Sub. Con | I2V Back Con. | Video Quality (w/ D.D.) | Video Quality (w/o D.D.) | Vbench Avg. |
|---|---|---|---|---|---|---|---|---|
| Low and Medium Resolution | | | | | | | | |
| ConsistI2V (Ren et al., 2024) | [256×256] | ✓ | 7.63 | 95.82 | 95.95 | 78.87 | 85.74 | 88.27 |
| DynamicCrafter (Xing et al., 2024b) | [256×256] | ✓ | - | 97.05 | 97.56 | 80.18 | 85.00 | 88.07 |
| DynamicCrafter (Xing et al., 2024b) | [512×320] | ✓ | 4.88 | 97.21 | 97.40 | 81.63 | 85.39 | 88.37 |
| SEINE (Chen et al., 2023) | [512×320] | ✓ | - | 96.57 | 96.80 | 79.49 | 85.71 | 88.45 |
| VideoCrafter (Chen et al., 2024a) | [512×320] | ✓ | 9.43 | 91.17 | 91.31 | 81.34 | 87.55 | 88.47 |
| CogVideoX Yang et al. (2024) | [720×480] | ✓ | 5.24 | 97.67 | **98.76** | 76.01 | 83.90 | 88.00 |
| AnimateDiff Guo et al. (2023) | [512×512] | ✓ | 1.50 | 84.21 | 84.48 | 70.80 | 78.92 | 81.98 |
| SEINE (Chen et al., 2023) | [512×512] | ✓ | 5.13 | 97.15 | 96.94 | 80.58 | 87.13 | 89.61 |
| Animate-Anything (Dai et al., 2023b) | [512×512] | ✓ | 1.58 | 98.76 | 98.58 | 81.21 | **88.84** | **91.30** |
| MarDini-L/ST-9 | [512×512] | ✗ | 2.24 | 98.64 | 97.12 | 80.84 | 88.22 | 90.64 |
| MarDini-S/ST-9 | [512×512] | ✗ | 2.24 | **99.04** | 97.23 | 81.00 | 88.59 | 90.98 |
| MarDini-L/T-17 | [512×512] | ✗ | 0.48 | 98.23 | 97.01 | 80.25 | 87.68 | 90.16 |
| MarDini-S/T-17 | [512×512] | ✗ | **0.46** | 98.76 | 97.18 | 80.56 | 88.17 | 90.62 |
| High Resolution | | | | | | | | |
| SVD-XT-1.0 (Blattmann et al., 2023a) | [1024×576] | ✓ | 2.19 | 97.52 | 97.63 | 82.79 | 86.54 | 89.30 |
| SVD-XT-1.1 (Blattmann et al., 2023a) | [1024×576] | ✓ | 2.19 | 97.51 | 97.62 | 82.23 | 86.66 | 89.38 |
| I2VGen-XL (Zhang et al., 2023b) | [1280×720] | ✓ | 6.01 | 96.48 | 96.83 | 81.17 | 87.02 | 89.43 |
| DynamiCrafter (Xing et al., 2024b) | [1024×576] | ✓ | 7.13 | 98.17 | **98.60** | 82.52 | 87.31 | 90.08 |
| MarDini-L/T-17 | [768×768] | ✗ | 1.01 | 98.34 | 96.63 | 80.88 | 88.22 | 90.54 |
| MarDini-S/T-17 | [768×768] | ✗ | **0.98** | 98.77 | 96.78 | 81.29 | 88.68 | 90.95 |
| MarDini-L/T-17 | [1024×1024] | ✗ | 1.80 | 98.61 | 96.34 | 81.35 | 88.69 | 90.89 |
| MarDini-S/T-17 | [1024×1024] | ✗ | 1.77 | **98.78** | 96.46 | 81.74 | **88.97** | **91.13** |

evaluation metrics. While, we observe notable benefits from scaling up the MAR model: MarDini-L excels in video interpolation and aligns closely with physical principles in image-to-video generation. To substantiate this, we conducted a user study involving 10 participants who evaluated 20 video pairs for physical accuracy. The study reveals that MarDini-L significantly outperforms Mardini-S, with a preference rate of 85.83%, indicating a clear preference for MarDini-L's outputs.

### 3.4 Additional Applications

In addition to image animation and video interpolation, MarDini also demonstrates potential for several other interesting applications, including zero-shot 3D view synthesis, video expansion, and hierarchical auto-regressive generation for slow-mo videos. Due to space constraints, please refer to Appendix G for details.

## 4 Related Work

**Auto-Regressive Model in Visual Generation.** Auto-regressive (AR) models (Gers et al., 2000; Hochreiter & Schmidhuber, 1997; Schmidhuber, 2015) have proven effective in natural language modeling (Brown, 2020; Achiam et al., 2023; Dubey et al., 2024; Team et al., 2023). To adapt this scalable modeling strategy for image and video generation, recent approaches (Yu et al., 2024; Chang et al., 2022; Li et al., 2023a; Yu et al., 2023a; Chang et al., 2023; Yu et al., 2023a) replace causal attention in AR with bidirectional attention, allowing for better capture of dense relationships in visual space.

Many studies (Yu et al., 2023b; Chang et al., 2023; Team, 2024; Xie et al., 2024) validate the scalability of this approach. To align with the training recipes from LLMs, these studies adopt discrete visual representations, using image tokenizers (Esser et al., 2021; Yu et al., 2021; Van Den Oord et al., 2017) to quantize continuous pixel values into discrete representations. The MAGViT series Yu et al. (2023a; 2024) adopts this strategy for video generation, demonstrating its effectiveness and contributing to the success of VideoPoet Kondratyuk et al. (2023). However, Li et al. (2024); Ramesh et al. (2021); Razavi et al. (2019) argue that this strategy suffers from unstable training and may limit model capacity due to the inherently continuous nature of visual

data. This inspires recent works (Li et al., 2024; Zhou et al., 2024) to shift towards continuous latent spaces for masked auto-regressive models to address these limitations.

We follow this trajectory but diverges in two ways: i) We highlight the importance of mask ratios, which were fixed in earlier works Li et al. (2024). By dynamically adjusting them with a progressive training strategy, we improve both model scalability and stability. ii) We propose an asymmetric input resolution design, allowing MAR to be effectively trained with full-resolution inputs.

**Diffusion Model for Video Generation.** In recent years, diffusion models (Ho et al., 2020; Neal, 2001; Jarzynski, 1997) have become a leading approach for both image and video generation (Rombach et al., 2022; Dhariwal & Nichol, 2021; Ramesh et al., 2022; Chen et al., 2024c; Saharia et al., 2022; Brooks et al., 2024; Dai et al., 2023a; Girdhar et al., 2023; Menapace et al., 2024; Kondratyuk et al., 2023; Cong et al., 2024; Li et al., 2025). These models conceptualize the generation process as gradually refining a real sample from Gaussian noise, demonstrating significant scalability and stable training. This paper empirically focuses on two key video generation tasks: image-to-video generation (Ren et al., 2024; Xing et al., 2024b; Chen et al., 2023; 2024a; Guo et al., 2023; Dai et al., 2023b; Blattmann et al., 2023a; Zhang et al., 2023b; Xing et al., 2024b) and video interpolation (Wang et al., 2024c; Voleti et al., 2022; Guo et al., 2024; Dong et al., 2023; Li et al., 2023b; Huang et al., 2022; Reda et al., 2022; Danier et al., 2024; Jain et al., 2024; Xing et al., 2024a).

We offer two key insights into video generation: i) Previous methods (Wu et al., 2023; Ho et al., 2022; Zhang et al., 2023a; Blattmann et al., 2023b; Wang et al., 2023; Girdhar et al., 2023; Gao et al., 2024) often first pre-train an image generative model, and then fine-tune it for video generation, or they require joint training for both tasks (Chen et al., 2024c; Esser et al., 2023). Alternatively, some studies Guo et al. (2023); Cong et al. (2024); Khachatryan et al. (2023) adopt tuning-free (or without specific-tuning) approaches to adapt existing image-based models for text-to-video tasks. While multi-stage pre-training on diverse inputs can be beneficial, video generation is often limited by the success of image-based pre-training, which typically serves as a secondary task. This paper proposes an alternative: training video generation models from scratch with progressively increasing task complexity. ii) Previous research (Girdhar et al., 2023; Wang et al., 2023; Chen et al., 2024c; Blattmann et al., 2023b) has predominantly employed temporal attention mechanisms to capture temporal dependencies, mainly due to the high computational and memory costs associated with spatio-temporal attention. However, in alignment with previous work (Blattmann et al., 2023b; Gao et al., 2024) suggesting that spatio-temporal attention enables superior video modeling, we propose an amortized strategy that makes spatio-temporal attention computationally feasible, even at high resolutions.

Finally, our paper also relates to asymmetric neural networks. Due to page limitations, additional related works are discussed in Appendix E.

## 5 Conclusion

We have introduced a new family of generative models for video, i.e., MarDini, based on auto-regressive diffusion, wherein a large planning model offers powerful conditioning to a much smaller diffusion model. Our design philosophy considers efficiency from model conception, and so our heaviest model component is only executed once at lower resolution inputs, whereas our generative module focuses on fine-grained details at the frame level, reconciling high-level conditioning and image details. Our model is unique in that it leverages a masked auto-regressive loss directly at the frame level. MarDini is afforded with multiple generative capabilities from a single model, e.g., long-term video interpolation, video expansion, and image animation. Our investigation shows that our modeling strategy is powerful enough to obtain competitive results on various interpolation and animation benchmarks, while doing it at a lower computational needs than counterparts with comparable parameter size. We discuss the limitation of our study in Appendix 6.

## 6 Limitations and Future Works

**Post Training.** The primary goal of this paper is to demonstrate the feasibility and effectiveness of combining masked auto-regressive (MAR) models with diffusion models (DM) for video generation. Consequently, we allocated the majority of our computational resources to the pre-training stage, placing less emphasis on post-training, despite its recognized importance in generative models (Dai et al., 2023a; Dubey et al.,

2024; Touvron et al., 2023). Post-training will be a top priority in our future work, focusing on enhancing long-term planning, improving motion quality, and achieving higher resolutions.

**Improved Conditional Signals.** A significant contribution of this work is the exploration of training a video generation model without relying on generative image pre-training. However, this approach presents a trade-off: MarDini is not inherently equipped with a text encoder for processing textural instructions. To conserve computational resources and quickly validate the feasibility of our method, we intentionally excluded commonly used conditional signals, such as text embeddings and motion scores. Encouraged by the initial success of our model, we plan to incorporate these conditional signals into MarDini in our future updates to broaden its range of applications.

## Ethics Statement

This paper explores the theoretical foundations of neural architecture design for video generation, rather than being tied to specific commercial applications. Consequently, the potential negative impacts of Mar-Dini align with those of other video generation models and do not pose unique risks that require special consideration. Importantly, unlike previous models trained on web-scale data, which may raise concerns about data copyright, MarDini is exclusively trained on a licensed Shutterstock dataset, without having such conflicts.

## Reproducibility Statement

We ensure reproducibility by providing detailed model configurations in Appendix A, along with the complete training recipes outlined in Appendix B. However, due to organizational policies, the model was trained using internal infrastructure and proprietary dependencies that cannot be publicly released. Additionally, the VAE component is based on internal product data and was not developed within this project, further restricting the potential release of model weights.

## Acknowledgements

The authors thank Mingchen Zhuge, Jinheng Xie, Yuren Cong, Kam Woh Ng, Aditya Patel, and Jinjie Mai for their valuable suggestions and contributions to the paper review. Haozhe Liu and Jürgen Schmidhuber were supported by funding from the King Abdullah University of Science and Technology (KAUST) - Center of Excellence for Generative AI under award number 5940 and the SDAIA-KAUST Center of Excellence in Data Science and Artificial Intelligence.

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

# A  Model Configuration

As outlined in Table 5, this study develops four models with distinct configurations. We train two planning models with 3.1B and 1.3B parameters alongside two generation models, employing spatio-temporal or temporal attention mechanisms. To align with our asymmetric design between the planning and generation models, the generation model's parameter size is reduced to $3\times$ or $10\times$ smaller than that of the planning model. Due to the high computational cost of spatio-temporal attention, we limit MarDini-L/ST and MarDini-S/ST to a 9-frame length for fair comparison on VIDIM-Bench (Jain et al., 2024). Importantly, the model's ability to autoregressively generate samples ensures that the length of the output video is not constrained.

| Configuration | Planning Model (MAR) | | | | | Generation Model (DM) | | | | | Frame |
| --- | --- | --- | --- | --- | --- | --- | --- | --- | --- | --- | --- |
| | Depth | Hidden Size | MLP Size | Attn. | Param. | Depth | Hidden Size | MLP Size | Attn. | Param. | |
| MarDini-S/ST | 8 | 4096 | 4096 | S.-T. Attn. | 1.3B | 8 | 1024 | 4096 | S.-T. Attn. | 288M | 9 |
| MarDini-L/ST | 16 | 4096 | 8192 | S.-T. Attn. | 3.1B | 8 | 1024 | 4096 | S.-T. Attn. | 288M | 9 |
| MarDini-S/T | 8 | 4096 | 4096 | S.-T. Attn. | 1.3B | 8 | 1024 | 4096 | T. Attn. | 288M | 17 |
| MarDini-L/T | 16 | 4096 | 8192 | S.-T. Attn. | 3.1B | 8 | 1024 | 4096 | T. Attn. | 288M | 17 |

Table 5: **Configuration Details of MarDini Models.** We provide four models, differing primarily in the size of the planning module (3.1B vs. 1.3B parameters) and the attention mechanisms used in the generation module: spatio-temporal attention (S.-T. Attn.) vs. temporal attention (T. Attn.). Here, we selected the parameter size of our generation model to align with the typical design of ViT-L (Dosovitskiy et al., 2020), as it has demonstrated excellent performance in frame-level generation tasks.(Wei et al., 2023).

# B  MarDini Training Manual

In Figure 6, we present the training details of MarDini. All experiments, including model variations, ablation studies, benchmark evaluations, and full model training, are carried out on a distributed MAST scheduler (Choudhury et al., 2024) using 256 H100 GPUs. The training dataset comprises approximately 34 million filtered Shutterstock videos, segmented into 2-second training clips. We use the AdamW optimizer for each stage with a $1.4 \times 10^{-4}$ learning rate and cosine learning rate scheduler. We adapt our batch size based on the resolution and the frame count to maximize GPU utility. For example, at $[256 \times 256]$ resolution with 9 frames, the batch size is 1024, processing 9K frames per iteration; at $[512 \times 512]$ resolution with 9 frames, the batch size is 720, processing 6480 frames per iteration. During inference, we set the classifier-free guidance (CFG)(Ho & Salimans, 2022) scale as 2.5 for the image-to-video task with the noise solver DDIM (Song et al., 2021), and we directly remove classifier-free guidance for video interpolation as it is redundant. FSDP (Zhao et al., 2023) and activation checkpointing (Zhao et al., 2023) are enabled to further save GPU memory. We do not include dynamic resolution training in our main training stages, as it slows down training. Instead, we find that after convergence, fine-tuning the model for a few steps (10K-20K) with dynamic resolutions enables it to quickly support this capabilities.

# C  Visualization of Video Interpolation

In Figure 7, we provide visualization results that demonstrate the superiority of MarDini in large motion modelling, compared to FILM (Reda et al., 2022), LDMVFI (Danier et al., 2024), and VIDIM (Jain et al., 2024).

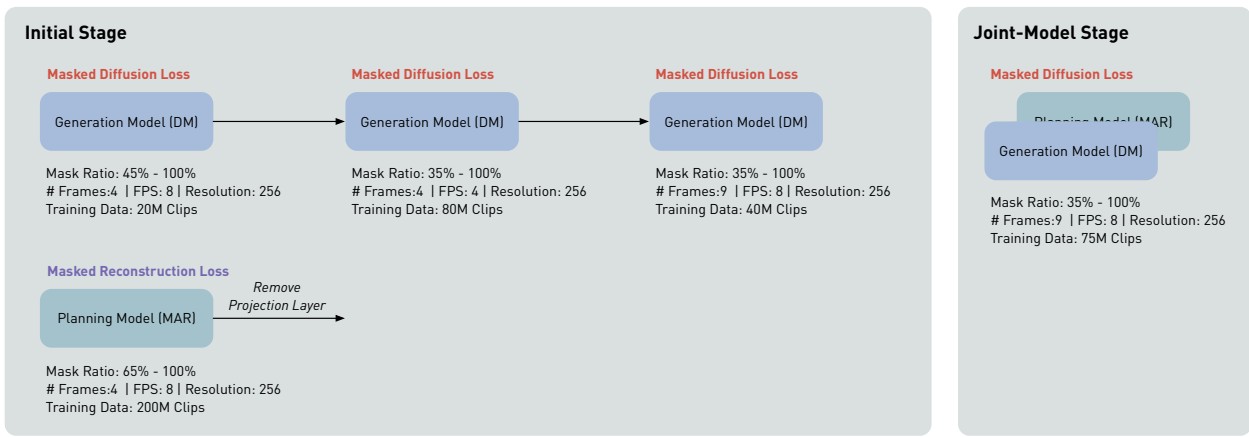

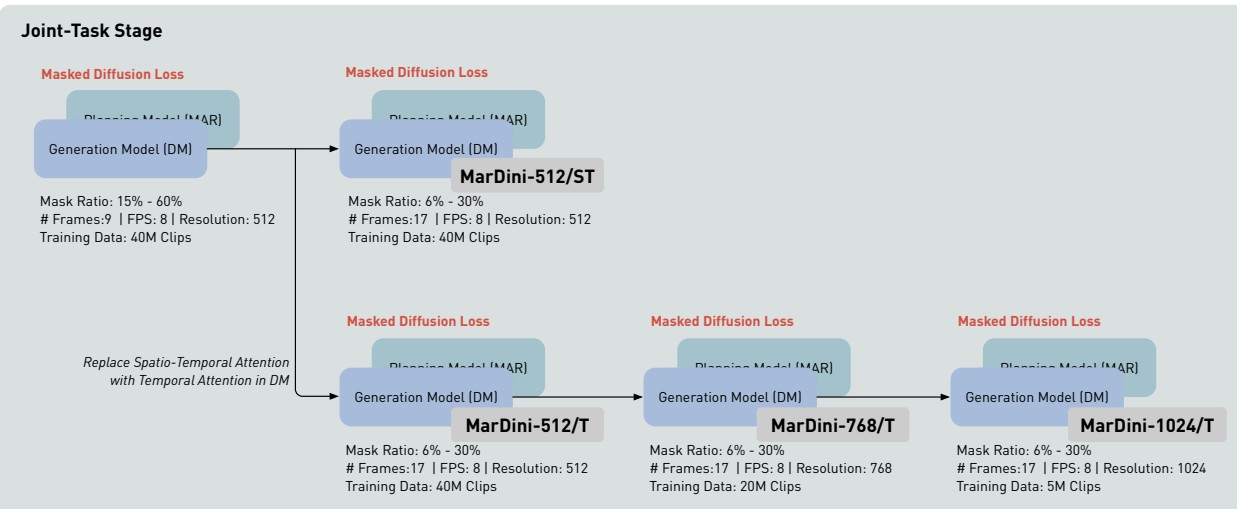

Figure 6: **MarDini Training Manual.** We list the mask ratios, frame rate (FPS), number of frames, and the size of training data for each training stage. Note that the total training data refers to the amount of data observed by the model for gradient updates, rather than the vanilla size of the training dataset. Our final model checkpoints are highlighted in gray. Mask Ratio refers to the proportion of frames preserved during training.

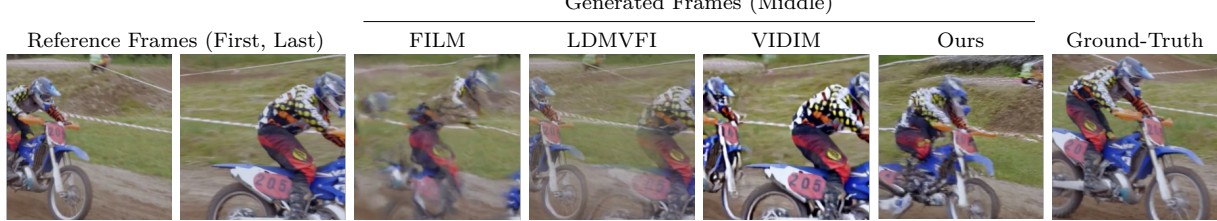

Figure 7: **Visualization of video interpolation methods conditioned on the first and last frames.** We present the generated frames from FILM (Reda et al., 2022), LDMVFI (Danier et al., 2024), VIDIM (Jain et al., 2024), and MarDini. The comparison results for these methods are sourced from Jain et al. (2024). We have included additional samples in the supplementary materials.

## D    Benchmarks

We evaluate the interpolation performance on VIDIM-Bench (Jain et al., 2024) and assess image animation performance on VBench (Huang et al., 2024).

For VIDIM-Bench, the task involves generating seven intermediate frames, with the first and last frames provided as conditions. The dataset includes approximately 400 videos from both DAVIS (Pont-Tuset et al., 2017) and UCF-101 (Soomro et al., 2012). We use FVD (Unterthiner et al., 2018) and FID (Heusel et al., 2017) as generation metrics, while adopting SSIM (Wang et al., 2004) and LPIPS (Zhang et al., 2018) as reconstruction metrics. Notably, we evaluate the middle (5th) frame for reconstruction metrics, as it presents the greatest challenge due to its distance from the reference frames.

For VBench, we utilize the official dataset to assess the model across several metrics: I2V-Subject Consistency, I2V-Background Consistency, and video quality. The video quality evaluation considers dimensions such as Subject Consistency, Background Consistency, Smoothness, Aesthetic Score, Imaging Quality, Temporal Flickering, and Dynamic Degree. Given that our model lacks text supervision, we omit the evaluation for video-text camera motion. Furthermore, since our model is pre-trained without incorporating dynamic degree guidance (known as motion score/strength), it is not directly comparable with other models in this respect. Therefore, we additionally report video quality by averaging all the dimensions except for Dynamic Degree and provide the VBench average score derived from I2V-Subject Consistency, I2V-Background Consistency, and the video quality dimensions (excluding dynamic degree). For the latency analysis, we ensure fairness by using the same computational platform: a single Nvidia A100 80G GPU. All implementations are based on their official code without any engineering optimizations. For MarDini, we simply employ `bf16` mixed precision to enhance computational efficiency. To account for variations in frame number and resolution, all results are normalized by frame count and evaluated at a consistent resolution of either [512 × 512] or [768 × 768].

## E  Additional Related Works

This paper also relates to asymmetric neural architectures, widely used in neural networks since the 1990s (Schmidhuber, 1992a;b). In computer vision, to achieve high-resolution generation, many studies (Podell et al., 2023; Pernias et al., 2024; Saharia et al., 2022; Li et al., 2024; Jain et al., 2024; Kang et al., 2023) employ a common strategy: a model generates low-resolution/quality samples, followed by another model that performs super-resolution (Kang et al., 2023), refinement (Podell et al., 2023), or interpolation (Wang et al., 2024b) to enhance the generation quality. In discriminative video models, asymmetric training strategies have been used for temporal segmentation models, where the full temporal extension does not fit the available GPU memory Xu et al. (2021). Since computational costs are distributed across stages, this approach is well-supported by existing computational platforms. Building on this trajectory but extending beyond it, we propose a novel design that partitions the model into two distinct models: a planning model and a generation model. The planning model, containing the majority of the model's parameters, is trained auto-regressively at a low resolution to generate conditional signals without producing visual outputs. These signals are then processed by the lightweight generation model, which converts them into high-resolution visual outputs using a diffusion process.

Unlike the traditional auto-regressive diffusion model (Li et al., 2024), which still faces high computational costs as resolution increases, we use cross-attention as an information pathway to connect asymmetric resolution input for more efficient training/inference.

## F  Discussions on DM Size

Based on recent findings (Liang et al., 2024), which demonstrated that pre-training loss is closely correlated with final generation performance, we adopt pre-training loss as a proxy metric to evaluate model performance. In this study, we fixed our pre-trained 3B MAR model as the planning component and

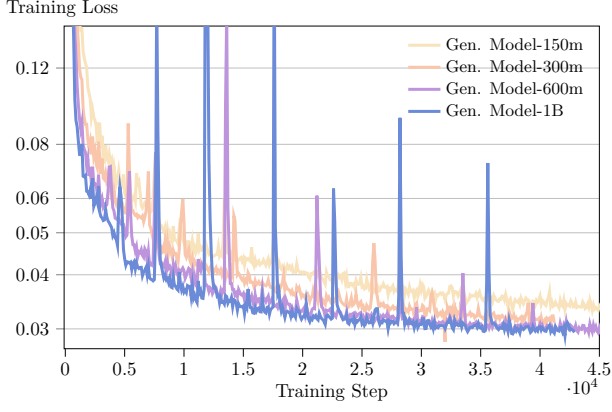

Figure 8: **Training Loss by Model Size.** We consider four DM sizes, ranging from 150M to 1B parameters.

trained different DM model sizes using the same filtered Shutterstock dataset. All models were trained under identical configurations, including a batch size of 512, video clips of 4 frames, a resolution of 256, and a mask ratio ranging from 0.65 to 1.00. Training was conducted on 16 H100 GPUs and evaluated at the same selected training steps in each training configuration. Consistent with prior studies, larger DM models deliver superior performance but at the cost of increased latency and training instabilities. Notably, the 150M DM model is a bit too small, showing persistently higher loss values even at the later stage of training. The 300M DM model (used in our paper) achieves a balanced trade-off, providing performance comparable to larger models while significantly reducing latency and training instabilities. For applications with a larger inference budget, increasing the DM model size could further improve performance, though with diminishing returns at larger scales.

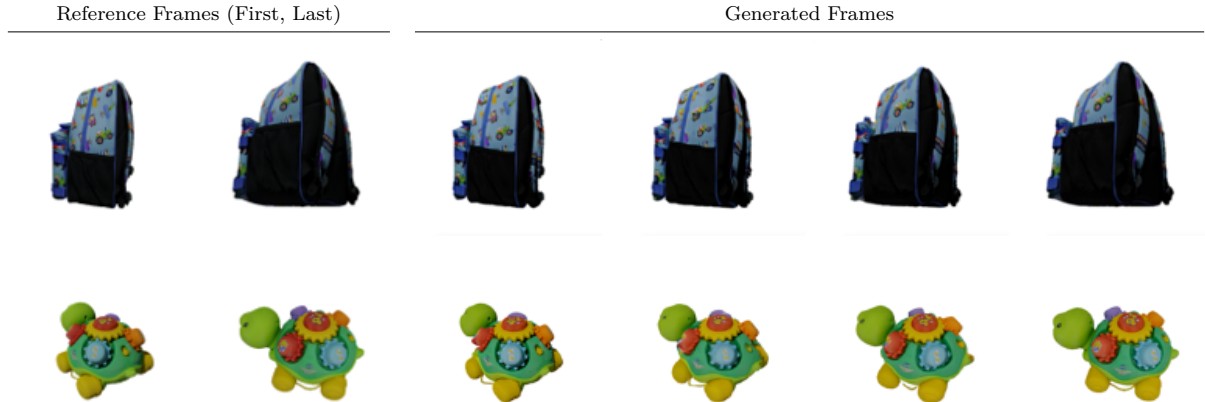

Figure 9: **Visualization of novel view synthesis conditioned on the two views.** Starting with two views of an object, MarDini generates the intermediate "frames", creating novel views. Notably, MarDini is trained without any 3D data but still manages to capture spatial information through video. The data is sourced from public research data (Downs et al., 2022).

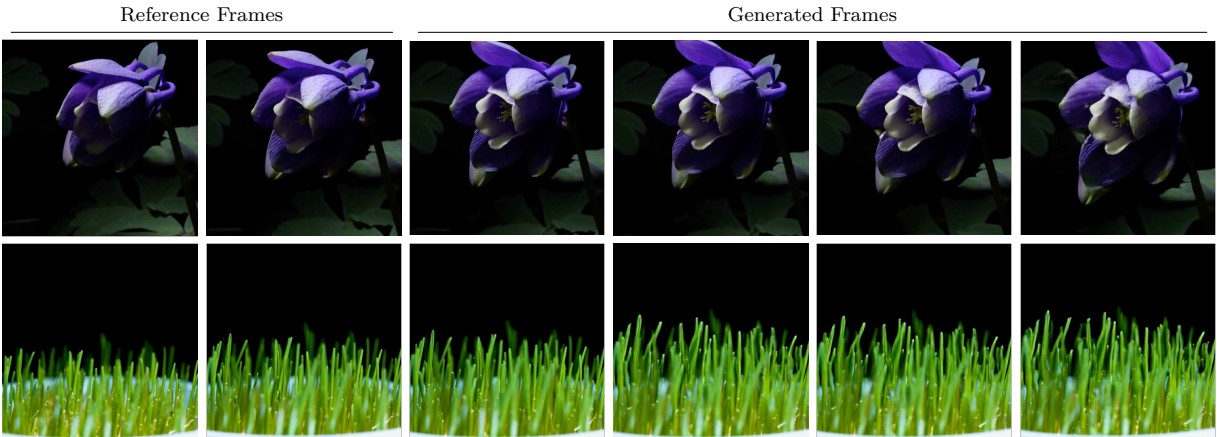

Figure 10: **Visualization of Video Expansion.** The model is conditioned on a sequence of 16 consecutive frames to predict the subsequent 12 frames. The video data used for visualization is sourced from public research data (Nan et al., 2024).

## G  Other Applications

MarDini is a novel video generation model that unifies MAR and DM into a single framework. This study reveals key insights and identifies specific weaknesses in this prototype, guiding our future research directions.

**Zero-Shot 3D View Synthesis** We demonstrate MarDini's potential for 3D view synthesis. Although trained solely on video data, MarDini shows preliminary spatial understanding, suggesting its possibilities for 3D applications. In Figure 9, two views of a fixed object serve as the first and last reference frames, while intermediate frames are generated, as in our video interpolation task. The model effectively generates convincing 3D-consistent views, highlighting its promising potential for 3D generation. Notably, no camera motion control signals are used, and we will explore MarDini on 3D data with better control in the future work.

**Video Expansion** MarDini integrates many of MAR's advantages, including the support for video expansion, where the conditional input is a set of frames rather than a single image. In this setup, motion information is implicitly embedded in the input. As shown in Figure 10, MarDini can effectively predict video sequences based on the provided motion cues (e.g., flower blooming, grass growing).

**(Hierarchical) Auto-Regressive Generation** By utilizing MAR for high-level planning, MarDini also supports auto-regressive inference, generating more frames beyond the one defined in the training stage. We demonstrate this through hierarchical auto-regressive generation: starting with a given video, we segment it into multiple clips, expand each clip segment, and treat the expanded clip segment as the new video for recursive video interpolation. In supplementary material, we provide some examples where, starting with 4 images, MarDini with a 17-frame window size auto-regressively expands them into a 256-frame video (64× expansion). This illustrates that our model is not limited by the training window size, highlighting its potential for long-range video generation.

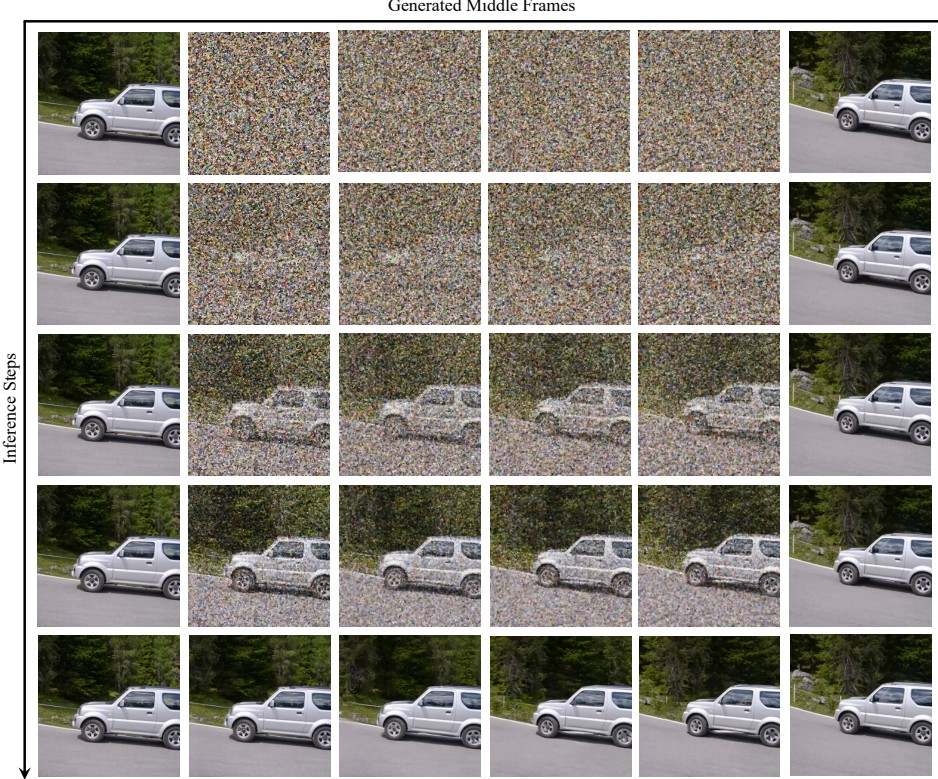

Figure 11: **Generated Frames Across Inference Steps.** The model generates intermediate frames between the first and last conditions. Columns show frames 1, 4, 8, 12, and 17; rows show results after 1, 5, 10, 15, and 25 steps.

## H    Additional Visualization

To better understand our model's behavior, we present generated frames at different inference steps. As shown in Fig. 11, the model achieves reasonably accurate motion after just 10 of the 25 total steps. In subsequent steps, it progressively refines frame details. Since the results in Fig. 4 suggest that motion is primarily guided by the planning model, we believe this explains why satisfactory results are achieved within 10 steps (see Fig. 5a). The planning model effectively outlines the motion in advance, simplifying the diffusion process.

