# OpenReview forum: "MarDini: Masked Auto-regressive Diffusion for Video Generation at Scale"
_TMLR — Accepted by TMLR_

### Review · Reviewer_dbvZ · 2025-02-19

**Summary Of Contributions:**

This paper presents **MarDini**, a video diffusion model that integrates masked auto-regression (MAR) within a unified diffusion framework, where MAR handles temporal planning and diffusion focuses on spatial generation. MarDini follows an asymmetric design: a MAR-based planning model generates low-resolution planning signals, while a lightweight diffusion model refines high-resolution frames. This flexible approach enables a single model to handle video interpolation, image-to-video generation, and video expansion by conditioning on masked frames at any positions. With its efficient resource allocation, MarDini achieves state-of-the-art video interpolation and generates high-quality videos in just a few inference steps, rivaling more expensive models.

**Audience:**

Yes

**Claims And Evidence:**

No

**Requested Changes:**

**Critical Changes (Necessary for Acceptance)**

1. **Clarify the Motivation and Claims about Training Instability**
+ The paper states that discrete representations cause training instability, yet no evidence is provided.
+ If this claim is central to the motivation, the authors should include empirical results demonstrating such instability.
+ If the claim is not well-supported, it should be revised or removed to avoid misleading conclusions.
2. **Provide Justification for the Masking Strategy**
+ The paper suggests that gradually increasing the masking ratio improves performance, but no ablation study is presented.
+ The authors should conduct an ablation study comparing different masking strategies, such as fixed-ratio random masking, to justify this design choice.
3. **Improve Empirical Comparisons**
+ The paper does not sufficiently demonstrate the superiority of the proposed method.
+ A comprehensive efficiency comparison should be added, including:
    - Inference speed with baseline mentioned below (item 5)
    - Model parameters
    - Memory usage
+ The results should be benchmarked against strong baselines such as MAGVIT v1, v2 and AnimateDiff to ensure a fair assessment.
4. **Revise Figure 1 for Accuracy**
+ The visual depiction of the VAE encoder is misleading, as it suggests that the output is an image rather than a feature map.
+ The figure should be corrected to accurately reflect the nature of the latent representation.
5. **Expand Related Work and Comparative Discussion**
+ Several relevant works in image-to-video and video interpolation are missing.
+ The authors should include comparisons with the following models:
    - AnimateDiff (ICLR 2024)
    - MAGVIT v1, v2 (CVPR 2023, ICLR 2024)
+ A discussion of how the proposed method differs from or improves upon these approaches is necessary.
---
**Suggested Changes (Would Strengthen the Paper)**

6. **Refine the Clarity of Section 2 ("Scalability")**
+ The paper claims that the proposed method does not require image-based pretraining, but it does not explain why this is an advantage.
+ Since many powerful self-supervised models exist, the authors should clarify whether their method can integrate pretraining or if it offers a trade-off between scalability and performance.
7. **Simplify the Training Pipeline if Possible**
+ The method introduces a multi-stage training process with separate transformer-based models, making it more complex than many existing approaches.
+ If an alternative, more streamlined training paradigm exists, the authors should discuss or explore it.
8. **Improve Writing Clarity and Technical Accuracy**
+ Some statements are overgeneralized or imprecise, particularly in the introduction and motivation.
+ The discussion of Auto-Regressive (AR) transformers should be revised, ensuring that references are appropriately aligned with the context.
9. **Additional Visualizations for Model Interpretability**
+ Since the method integrates masked auto-regression and diffusion, it would be beneficial to include visualizations of:
    - How planning signals evolve over time in the MAR component.
    - Intermediate frames generated by the diffusion process to illustrate the model’s behavior.
10. **Ablation on Model Components**
+ The model combines masked generative transformers and diffusion models, but it is unclear how much each contributes to performance.

---
**Summary of Key Revisions**

**Critical:** Justify training instability claims, add masking ablations, expand empirical comparisons, revise Figure 1, and include missing related works.

**Suggested:** Clarify scalability, simplify training if possible, refine writing, improve visualizations, and conduct additional ablations.

**Strengths And Weaknesses:**

**Strengths:**

+ The paper is well-written and presents a complete exposition of the proposed method.
+ The applications are interesting and relevant to the field.
+ The idea of integrating masked generative transformers with diffusion models is promising.

**Weaknesses:**

+ **Clarity Issues:** The paper states, "Auto-regressive (AR) transformers... sparking efforts to achieve similar breakthroughs in computer vision (Rombach et al., 2022; Dai et al., 2023a; Saharia et al., 2022)," but Rombach et al., 2022 (a U-Net-based latent diffusion model) does not seem directly relevant to the AR mechanism mentioned in this context.
+ **Questionable Motivation:** The paper claims that training instability due to reliance on discrete representations is a key bottleneck, yet no evidence is provided to support this. Based on my experience with masked generative transformers, including MaskGIT and its variants, I have not observed such instability—these models use cross-entropy loss, which is generally stable. This weakens the motivation for the proposed approach.
+ **Scalability Claims:** The paper highlights the advantage of not requiring image-based pretraining. However, given the availability of strong foundation models trained on large-scale data in a self-supervised manner, incorporating such pretraining could be beneficial rather than restrictive. Thus, this contribution is not as significant as suggested.
+ **Figure 1 Confusion:** The visualized image before and after the VAE encoder appears identical, which is misleading. Since VAE outputs feature maps rather than reconstructed images, the figure should be revised for clarity.
+ **Increased Complexity:** The proposed method is more complex than existing approaches, requiring separate model configurations for Transformers and a multi-stage training process that is not end-to-end. This reduces both flexibility and scalability.
+ **Lack of Ablation Studies:** The paper states that increasing the masking ratio creates more challenging generation tasks but provides no ablation study to justify this design choice. A comparison with simpler masking strategies (e.g., fixed or randomly sampled masks) is needed to assess its impact on performance.
+ **Weak Empirical Performance:** The proposed method performs worse on multiple tasks/datasets, failing to convincingly demonstrate its superiority. Moreover, compared to masked non-auto-regressive models like MAGVITv2, its efficiency remains questionable. A thorough comparison of model parameters, inference speed, and memory usage is needed to justify its advantages over existing approaches.
+ **Missing Comparisons:** Several important works in image-to-video and video interpolation, such as AnimateDiff [1] and MAGVIT v1, v2 [2,3], are missing from the comparisons. Their inclusion would provide a more comprehensive evaluation.

**Overall Assessment:**

While the paper presents an interesting combination of masked generative transformers and diffusion models, its advantages remain unconvincing due to weak motivation, increased complexity, and lack of empirical support. A more thorough justification, additional ablations, and stronger comparisons with existing methods are needed to clarify its contributions.

**References:**

[1] AnimateDiff: Animate Your Personalized Text-to-Image Diffusion Models without Specific Tuning, ICLR 2024

[2] MAGVIT: Masked Generative Video Transformer, CVPR 2023

[3] Language Model Beats Diffusion -- Tokenizer is Key to Visual Generation, ICLR 2024

---

> ### Author Response · Authors · 2025-03-27
> **Response to Reviewer dbvZ**
>
> ## Motivation for dropping the discrete tokenization
>
> Thanks for pointing this out. The conclusion on "training instability" primarily stems from paper [A], which states: "Vector-quantized tokenizers are difficult to train and sensitive to gradient approximation strategies." To strengthen our claim, we will add an additional citation.
>
> Beyond training instability, our key motivation is that visual data consists of continuous pixel signals distributed across a high-dimensional space, making it naturally suited for continual latent representations. Recently, NVIDIA introduced the Cosmos World Foundation Model [B], which integrates both discrete and continual compressors for visual generation. With identical compression rates, model architectures, and parameter sizes, Cosmos provides a fair platform for evaluating continual vs. discrete representations. We leverage its pre-trained model and assess its reconstruction performance across multiple datasets (see Table R1). Since Cosmos is an actively evolving project, our results are based on the latest version as of March 2, 2025.
>
> Our empirical study demonstrates the advantages of continual latent space, achieving superior performance in various ablations. As the performance of the VAE (tokenizer) defines the upper bound of a generation model, we believe MarDini's motivation is well addressed. We have revised our paper's motivation section using orange text for clarity and emphasis.
>
> Regarding the comparison with MAGVIT, we'd like to admit that a direct and fair comparison is quite challenging, considering that their model weights (and code for MAGVIT-v2) are not publicly available [C]. Meanwhile, the MAGViT model series is not trained on web-scale data, making fair comparisons even more difficult, particularly in zero-shot evaluation scenarios. However, given that VAE-based approaches outperform tokenizers under a restricted experimental setting, we believe MarDini follows a more promising trajectory. We have discussed it in our updated related works.
>
> Beyond empirical studies, this work offers orthogonal insights into the MAGViT series in two key aspects: 1) MarDini introduces a novel video generation framework that combines MAR (based on discrete codes) with diffusion models (based on continuous latent codes), leveraging MAR's flexibility and the stronger open-source support of continuous VAEs and diffusion models [D-F].  2) Its decompositional design naturally enables an asynchronous resolution input strategy, enhancing training and inference efficiency.
>
> TABLE R1 The reconstruction performance of Cosmos compressors on different datasets. We consider two different types of data: video and image. The compression rate is 8 times on the temporal dimension and 8 times on the spatial dimension.
> | Dataset         | PSNR (CV) | SSIM (CV) | LPIPS (CV) | PSNR (DV) | SSIM (DV) | LPIPS (DV) |
> |-----------------|-------------:|-------------:|--------------:|-------------:|-------------:|--------------:|
> | imagenet-val    |       31.092 |        0.920 |         0.041 |       28.026 |        0.837 |         0.135 |
> | text-ocr         |       33.679 |        0.964 |         0.013 |       30.663 |        0.927 |         0.038 |
> | **Image Average** | **32.385** | **0.942** | **0.027** | **29.344** | **0.882** | **0.086** |
> | BDD100K         |       34.360 |        0.950 |         0.020 |       27.709 |        0.854 |         0.116 |
> | bridgedata-v2   |       34.654 |        0.953 |         0.017 |       27.881 |        0.874 |         0.092 |
> | ego-exo-4d      |       33.775 |        0.952 |         0.022 |       27.027 |        0.844 |         0.148 |
> | panda-70m       |       34.737 |        0.967 |         0.018 |       26.859 |        0.883 |         0.103 |
> | **Video Average** | **34.382** | **0.956** | **0.019** | **27.369** | **0.855** | **0.124** |
>
>
> [A] Li, Tianhong, et al. "Autoregressive image generation without vector quantization." Advances in Neural Information Processing Systems 37 (2025): 56424-56445.
>
> [B] Agarwal, Niket, et al. "Cosmos world foundation model platform for physical ai." arXiv preprint arXiv:2501.03575 (2025).
>
> [C] https://github.com/google-research/magvit/issues/16
>
> [D] https://github.com/Tencent/HunyuanVideo
>
> [E] https://github.com/Wan-Video/Wan2.1
>
> [F] https://github.com/stepfun-ai/Step-Video-T2V

---

> ### Author Response · Authors · 2025-03-27
> **Response to Reviewer dbvZ**
>
> ## Ablation study on Mask Ratios
> Thanks for the suggestion.
> Here, we conduct two ablation studies:
>
> 1) Fixed Drop Ratio Training: In the first experiment, we train the model from scratch using a fixed drop ratio of 0.5 to generate 4-frame videos. Since the model never encounters sequences conditioned on a single image, it fails to perform image-to-video generation. Consequently, we omit numerical results for this setting.
>
> 2) Mask Ratio Variation: Starting from a mid-training checkpoint, we continue training the model with two approaches: using a fixed mask ratio of 0.5 and randomly sampling a mask ratio between 0 and 1. The table below presents their performance on VBench, which falls short of our model trained with a progressive mask ratio strategy. Each model is trained for 20k steps at a resolution of 768 using 16 A100 GPUs.
>
> | **Mask Ratios**                | **VBench (Avg.)** |
> |---------------------------------|-------------------|
> | Fixed Mask Ratio (0.5)          | 89.29             |
> | Random Mask Ratio (0-1)         | 89.42             |
> | Progressive Mask Ratio          | **90.54**             |
>
> ## Comparison with the AnimateDiff
>
> We appreciate the reviewer's suggestion and have included AnimateDiff in the related works. Compared to AnimateDiff and other diffusion-based models, our framework offers greater flexibility. Previous diffusion models are typically trained for specific tasks, such as image-to-video generation, and require additional fine-tuning for tasks like video interpolation or prediction. In contrast, our framework provides a unified solution for a wide range of video and image-to-video tasks within a single model without additional modifications.
>
> As suggested, we follow the default configuration of AnimateDiff on Vbench bench for image-to-video tasks, with the performance comparing with MarDini below,
>
> | Model          | Parameter Size | Memory Usage  | Latency | VBench Avg.  |
> |----------------|----------------|---------------|---------|--------------|
> | AnimateDiff    | 2.4B           | 23 GB         | 1.50    | 81.98        |
> | Ours           | 1.6B           | 22 GB         | 0.46    | 90.62        |
>
> The proposed model outperforms AnimateDiff in both efficiency and generation performance. We have also included this result in the experiment section.
>
> ## Pre-training on Image Data
> Thanks for pointing that out. We admit that image-based pre-training can improve performance. Here, we expect that MarDini can offer a key advantage in model verification. Traditional methods require training on image data followed by temporal modeling, which remains costly. This means testing a new model on video often demands prior adaptation for image generation. MarDini suggests that skipping this image pre-training step should be acceptable, which may enable faster prototype verification, potentially accelerating research in video generation. We have also revised the paper for clarification.

---

> ### Author Response · Authors · 2025-03-27
> **Response to Reviewer dbvZ**
>
> ## Additional visualization
> As suggested, we added intermediate frames generated at different timesteps to the paper. The visualization shows that after just 10 inference steps, the object's motion becomes evident. We also present results with and without the planning model in Fig. 4, which suggest that the planning model primarily guides motion. We thus reasonably suspect that the planning model (MAR) progressively injects motion guidance into the generation model, thereby accelerating the diffusion process. Since the planning model produces a highly compressed embedding code, visualizing its distribution across inference steps offers limited insight. We leave further analysis of the planning model's output for future work.
>
> ## Technical Accuracy and More Related Works
> As suggested, we made several revisions to the manuscript:
>
> 1) We reviewed and corrected the citation for the introduction.
> 2) We added AnimateDiff and MAGViT to the related works section and provided a detailed discussion of their contributions.
> 3) We expanded the related works section by including additional studies on video interpolation and image-to-video models.
> 4) We revised Figure 1’s caption to clarify that the model takes the latent code as input.
>
> ## Simplify the training pipeline
> Thanks for pointing this out. Following the suggestion, we have prioritized simplifying the training pipeline in our future research study.
>
>
> ## Ablation on Model Components
> We conducted experiments on the video interpolation task, the first video task our model addresses. The planning model has 3B parameters, while the generation model has 0.3B parameters. To enable the planning model to perform video interpolation, we introduced a projection layer that maps the embedding code to the video latent code using an MSE loss. Results show that while the smaller generation model outperforms the planning model alone, combining both achieves the best performance, highlighting the value of integrating MAR and DM models. We also conducted experiments with the 1B planning model, which yielded similar observations, as shown in Table 1.
>
> | **Planning Model** | **Generation Model** | **FVD (DAVIS)** | **FVD (UCF101)** |
> |---------------------|---------------------|------------------|-------------------|
> |     [Yes]           |   [No]              | 373.03           | 701.03            |
> |     [No]            |   [Yes]             |  320.89          | 383.04            |
> |     [Yes]           |   [Yes]             | **102.87**       | **197.69**        |

---

### Review · Reviewer_3oxA · 2025-03-11

**Summary Of Contributions:**

The paper introduces a generative model combining Masked Auto-Regressive planning and Diffusion Models for video synthesis. Its contributions include an asymmetric architecture using a heavy-weight MAR at low-resolution to guide a lightweight DM for high-resolution generation, resulting in flexible and efficient handling of tasks like video interpolation, image-to-video synthesis, and video expansion. An Identity Attention mechanism ensures stable training. Empirical results show state-of-the-art performance in video interpolation and efficiency in image-to-video generation without image-based pre-training.

**Audience:**

Yes

**Claims And Evidence:**

Yes

**Requested Changes:**

- Please conduct additional ablations on architectural choices, including varying MAR-to-DM parameter ratios and different resolutions, to justify the current design decisions.
- Please provide a deeper analysis of how Identity Attention stabilizes training, ideally with additional visualizations or quantitative evidence demonstrating convergence improvement.

**Strengths And Weaknesses:**

Strengths:
- It effectively combines masked autoregressive and diffusion models with an asymmetric architecture.
- It supports multiple tasks (video interpolation, image-to-video generation, and video expansion) within a unified framework.
- Achieves competitive performance with significantly reduced inference costs.
- Successfully transitions from simple interpolation to complex video generation, enabling training from scratch without image-based pre-training.
- Comprehensive experiments and ablation studies demonstrating model effectiveness.

Weaknesses:
- Lack of more analysis on alternative architecture designs (e.g., different resolutions or MAR-to-DM ratios).

---

> ### Author Response · Authors · 2025-03-27
> **Response to Reviewer 3oxA**
>
> ## Additional Ablation Study on Architectural Choices.
>
> ### Discussions on Resolution.
> In this paper, we trained MarDini on the same video data and ablated it with different input resolutions. The generation model operates at 1024 resolution, while the planning model uses 256 resolution. This choice addresses memory constraints; setting the planning model to 256 resolution allows a batch size of 1 when training the generation model at 1024 resolution. Increasing the planning model's resolution would exceed memory limits under commonly used memory-efficient training settings (e.g., with bf16 mixed precision and FSDP). While previous studies [A] highlight the performance benefits of higher resolutions, we adopt 256 resolutions for the planning model to maximize performance within these constraints. The memory analysis for these settings has been included in Table 2.
>
> [A] He, Kaiming, et al. "Masked autoencoders are scalable vision learners." Proceedings of the IEEE/CVF conference on computer vision and pattern recognition. 2022.
>
>
> ### Discussions on parameter size.
> For the parameter size, we acknowledge that both the generation and planning models benefit from parameter scaling. However, since the generation model operates at a higher resolution, scaling it is more computationally expensive. Thus, we design MarDini with a smaller generation model and a larger planning model. Below, we discuss their parameter designs separately.
>
> For the diffusion model's parameter size, we reference empirical findings from the study [B], which suggest that a model with approximately 300M parameters achieves relatively satisfactory performance on frame generation. Consequently, we adopt this parameter size for our diffusion model. To validate this assumption, we conduct empirical studies using pretraining loss as a proxy metric, following recent findings [C] that link pretraining loss to final generation performance. This approach allows evaluation within the limited time and budget of the rebuttal period. We fixed our pre-trained 3B MAR model as the planning component and trained various DM models on the same filtered Shutterstock dataset. All models used identical settings: batch size of 512, 4-frame video clips at 256 resolution, and a mask ratio from 0.65 to 1.00. The training was performed on 16 H100 GPUs and evaluated at the same training steps for each configuration.
>
> As shown in the table, larger DM models generally achieve better performance but at the cost of increased latency. The 150M DM model proves too small, exhibiting persistently higher loss values even in later training stages. In contrast, the 300M DM model, which we adopt in our paper, offers a balanced trade-off, delivering performance comparable to larger models while significantly reducing latency. For applications with greater inference budgets, increasing the DM model size may further improve performance. We have included this ablation in Appendix F.
>
> | Gen. Model |                 |   Pre-Training     |     Loss   |        | Latency (sec/frame) |
> |:----------:|:---------------:|:------:|:------:|:------:|:-------------------:|
> |            |       1.5k      |  2.5k  |  4.5k  |   20k  |                     |
> |    150M  |      0.1314     | 0.0908 | 0.0694 | 0.0403 |         0.26        |
> |    300M    |      0.0968     | 0.0773 | 0.0598 | 0.0360 |         0.48        |
> |    600M    |      0.0798     | 0.0649 | 0.0540 | 0.0332 |         0.97        |
> |     1B     |      0.0675     | 0.0587 | 0.0586 | 0.0319 |         2.15        |
>
>
> For the MAR model's parameter size, scaling up does not significantly increase the computational burden during inference or training, making it a more efficient approach for improving performance. In this paper, we train two MAR models: a 1B model (S) and a 3B model (L). The 3B model represents the largest feasible size for our 1024-level video generation task without complex infrastructure optimizations, given our use of bidirectional attention in each block.
>
> To assess the impact of scaling, we conducted a user study comparing the 1B and 3B models on image-to-video tasks. Ten participants evaluated 20 video pairs for physical accuracy, with results showing an 85.53% preference for MarDini-L's output. This highlights the clear benefits and promise of scaling the MAR model's parameter size.
>
>
> [B] Wei, Chen, et al. "Diffusion models as masked autoencoders." Proceedings of the IEEE/CVF International Conference on Computer Vision. 2023.
>
> [C] Liang, Zhengyang, et al. "Scaling laws for diffusion transformers." arXiv preprint arXiv:2410.08184 (2024).

---

> ### Author Response · Authors · 2025-03-27
> **Response to Reviewer 3oxA**
>
> ## Discussion on Identity Attention.
> We appreciate the reviewer's interest. We conducted an ablation study, shown in Figure 5(c), where the baseline model lacks Identity Attention. The results demonstrate that Identity Attention helps stabilize training.
>
> We have noticed that several concurrent studies have attempted to combine the auto-regressive approach with diffusion models, which all face this common challenge: distinguishing between conditional input tokens (frames) and noised inputs since they can appear at any position in the sequence. Existing solutions address this in different ways: 1) MARDiff [A] uses a mask indicator for the reference frame, 2) AR-Diffusion [B] employs the timestep as a signal, and 3) our approach leverages an attention mechanism to identify the reference frame.
>
> We agree with the reviewer that grouping conditional input tokens and noised tokens warrants further investigation. While several related studies [A, B] emerged around the time of our submission, we plan to explore this aspect in future work.
>
> Lastly, since the baseline model (without Identity Attention) failed to produce meaningful results (with Nan training losses), we were unfortunately unable to provide meaningful visualizations and numerical analysis.
>
> [A] Fan, Lijie, et al. "Scaling autoregressive text-to-image generative models with continuous tokens." The Thirteenth International Conference on Learning Representations. 2025.
>
> [B] Sun, Mingzhen, et al. "AR-Diffusion: Asynchronous Video Generation with Auto-Regressive Diffusion." arXiv preprint arXiv:2503.07418 (2025).

---

> > ### Comment · Reviewer_3oxA · 2025-04-14
> >
> > The responses provided adequately address my previously raised questions. Given the experimental results and clarified design decisions, I see no further issues requiring discussion from my side and recommend acceptance.

---

> > > ### Author Response · Authors · 2025-04-14
> > >
> > > We sincerely thank the reviewer for the time and effort contributed, and are pleased to see that our response has addressed the concerns raised.

---

### Review · Reviewer_9X6H · 2025-03-23

**Summary Of Contributions:**

This paper introduces MarDini, a video generation framework that combines Masked Auto-Regressive (MAR) modeling with Diffusion Models (DM) in an asymmetric architecture. The core idea is to use a MAR-based temporal planning module—operating on low-resolution inputs—for long-range sequence modeling, while a lightweight diffusion-based generator performs high-resolution spatial refinement via denoising. This design allows MarDini to support a wide range of video generation tasks—such as video interpolation, image-to-video generation, and video expansion—through a flexible frame-masking mechanism. The paper demonstrates that MarDini achieves state-of-the-art performance in video interpolation, while maintaining high efficiency during inference.

**Audience:**

Yes

**Broader Impact Concerns:**

As with other generative models, MarDini raises concerns regarding the potential misuse of synthesized video content (e.g., misinformation, deepfakes).

**Claims And Evidence:**

Yes

**Requested Changes:**

See weakness section in the last box.

**Strengths And Weaknesses:**

S:

* The combination of MAR and diffusion in a modular, asymmetric design is novel and addresses both temporal modeling and spatial generation efficiently.
* A single model is capable of performing various video generation tasks by simply varying the frame masking pattern, showcasing strong flexibility and generalization.
* MarDini achieves SOTA performance in video interpolation benchmarks and performs competitively in image-to-video generation, with significantly reduced inference cost compared to heavier diffusion-based baselines.

W:

* The current evaluations could be more comprehensive. For video interpolation, recent strong baselines like ToonCrafter should be included; for image-to-video generation, comparisons with CogVideoX would make the evaluation more convincing, especially considering their wide recognition and scaling benchmarks.
* Given the known limitations of VBench and perceptual metrics like FVD or IS, especially for nuanced tasks such as interpolation, a user study would significantly strengthen the empirical claims.
* An interesting aspect of MarDini is that it does not require pretraining on large-scale image models. This opens up the possibility of training the model from scratch on narrow-domain datasets. For instance, one could consider domain-specific applications like accident video synthesis (e.g., AVD2: Accident Video Diffusion for Accident Video Description, ICRA 2025). Demonstrating MarDini's effectiveness in such settings would underscore its practical flexibility and could inspire adoption beyond general video generation tasks.
* The hybridization of auto-regressive and diffusion paradigms is a thoughtful and technically elegant solution to combining long-range planning with high-quality generation. However, in the long term, it remains an open question whether hybrid architectures can scale as efficiently as cleaner, unified paradigms (e.g., fully AR or fully diffusion). The authors’ insights into this trade-off—especially in terms of scaling laws, inference efficiency, and training convergence—would be a valuable addition to the broader discussion on the future of general-purpose video models.

---

> ### Author Response · Authors · 2025-03-27
> **Response to Reviewer 9X6H**
>
> ## Compared with the additional models.
> Thanks for the valuable feedback. We have now added CogVideoX to our comparison. While CogVideoX demonstrates strong performance and broad recognition, MarDini currently does not support text guidance—a feature we have prioritized for future development. As a result, we focused on text-independent evaluation criteria. Our results indicate that MarDini performs comparably to CogVideoX, achieving superior I2V subject consistency (99.04 vs. 97.67) but slightly trailing in background consistency (97.23 vs. 98.76).
>
> ## User Studies on Video Interpolation
> Based on the suggestion, we conducted a user study comparing MarDini with ToonCrafter. The study involved 10 volunteers evaluating 20 videos covering cartoons, natural scenes, human subjects, and slow-motion content. Results show that MarDini outperforms ToonCrafter with a preference ratio of 72.5% to 27.5%. We have also updated our related works to include ToonCrafter. Since the benchmark model from VIDIM-Bench is not open-source, conducting a fair user study comparison with it remains challenging.
>
> ## Discussion on Parameter Scaling-Up
> Regarding the benefits of scaling, we empirically observe that both the Diffusion and MAR models improve with increased parameter sizes. Appendix F (and the table below) confirms that, with a fixed MAR model, the Diffusion model achieves better performance when scaled to larger parameters. Conversely, our ablation study demonstrates that enlarging the MAR model with a fixed Diffusion model better aligns predictions with underlying physical principles (a preference rate of 85.83% for larger models). Hence, combining these models preserves their scaling properties.
>
> Moreover, we highlight the asymmetric resolution design as a critical feature of our approach. Since the MAR model primarily handles long-term planning tasks, it does not necessitate high resolution. In contrast, the Diffusion model focuses on high-fidelity generation and thus benefits significantly from high-resolution inputs. From an efficiency standpoint, increasing MAR's parameter size is advantageous as it consistently processes fewer tokens.
>
> | Gen. Model |                 |   Pre-Training     |     Loss   |        | Latency (sec/frame) |
> |:----------:|:---------------:|:------:|:------:|:------:|:-------------------:|
> |            |       1.5k      |  2.5k  |  4.5k  |   20k  |                     |
> |    150M  |      0.1314     | 0.0908 | 0.0694 | 0.0403 |         0.26        |
> |    300M    |      0.0968     | 0.0773 | 0.0598 | 0.0360 |         0.48        |
> |    600M    |      0.0798     | 0.0649 | 0.0540 | 0.0332 |         0.97        |
> |     1B     |      0.0675     | 0.0587 | 0.0586 | 0.0319 |         2.15        |
>
>
>
> ## Train Model on Other Expert Domain
> We are encouraged by the reviewer's comment: "This opens up the possibility of training the model from scratch on narrow-domain datasets." We also agree on the value of pre-training our model on additional video datasets. We will cite the study [A] in related work. However, training on such datasets may need to be deferred to future work due to time constraints, particularly since dataset [A] was released after our submission.
>
> [A] Li, Cheng, et al. "AVD2: Accident Video Diffusion for Accident Video Description." arXiv preprint arXiv:2502.14801 (2025).

---

> > ### Comment · Reviewer_9X6H · 2025-04-04
> > **Open-source?**
> >
> > The additional comparisons with CogVideoX and ToonCrafter, as well as the user study and scaling analysis, are much appreciated and help strengthen the empirical claims of the paper. The clarification regarding domain-specific adaptability is also encouraging.
> >
> > Do the authors plan to release the code and model checkpoints, similar to CogVideoX, to facilitate downstream research? Open-sourcing MarDini would be particularly valuable for the community, especially for fine-tuning in domain-specific applications.

---

> > > ### Author Response · Authors · 2025-04-05
> > >
> > > We sincerely appreciate the reviewer’s time and effort. We are also glad to see that our responses have addressed the reviewer’s concerns.
> > > Regarding the open-source plan, we regret that we are unable to release the code and checkpoints at this time due to internal policies. However, to ensure reproducibility, we have provided full details of the training configurations, model architecture, and all relevant hyperparameters.

---

> > > > ### Comment · Action_Editor_nrrd · 2025-04-22
> > > > **About open-source plan**
> > > >
> > > > Dear authors,
> > > >
> > > > Thanks for your reply regarding the open-source plan of this work, and we understand that there might be policies that prevent you from releasing the models/checkpoints. But considering the potential impact of this work on the community and its benefit for following-up work, please could you be more specific about the *internal policies* that prevent you from open-sourcing if possible? And more details on the restrictions on open-sourcing the code, rather than the models or checkpoints.
> > > >
> > > > The open-sourcing of the code/implementation, if the models are impossible, would still be quite helpful and beneficial to the community.
> > > >
> > > > Please be reassured that the above is more of a suggestion and asking for more details instead of forcing you to open-source. The decision of this paper will not depend on whether it is open-sourced or not.
> > > >
> > > > Best,
> > > >
> > > > AE

---

> ### Author Response · Authors · 2025-04-23
>
> Dear AE,
>
> Thank you for your thoughtful message and for understanding the constraints around open-sourcing.
>
> To clarify, the model was trained using internal infrastructure and dependencies that are not permitted to be shared publicly due to organizational policy. This includes proprietary training frameworks and resource management tools that are deeply integrated into our development workflow.
>
> On the model side, the VAE component used in this work was not developed as part of this project. It is based on internal product data and subject to usage restrictions, which further complicates the possibility of releasing checkpoints or model weights.
>
> We genuinely appreciate the value of open-sourcing and understand how it can benefit the community. While we are currently unable to release the models or the full implementation, we will continue to explore whether a minimal and independent code release—excluding internal components—might be feasible.
>
> Thank you again for your support and for the opportunity to clarify.
>
> Best regards,
>
> MarDini Team

---

### Decision · Action_Editor_nrrd · 2025-05-01

**Recommendation:** Accept with minor revision

**Comment:**

In this paper, the authors presented a new group of diffusion models for video generation, by integrating the masked auto-regression (MAR) in a diffusion framework. The key idea and main contribution is around the MAR-based planning approach, together with the diffusion spatial generation. Experiments on video interpolation, image-to-video generation, and video expansion show the effectiveness and efficiency of the proposed model.

The paper was reviewed by three expert reviewers, followed by a discussion phase between the authors and reviewers, after which the major concerns from the reviewers were mostly addressed by the authors. The reviewers are satisfied with the authors' further response and clarifications, all recommending positive scores (3 Leaning Accept).
Considering the potential impact of this work, it would be great if the authors could release their models or implementation. The AE and reviewers understand the authors' concerns about open sourcing and acknowledged the authors' clarification on this. But the authors are suggested to add a paragraph to clarify this with details about the open-sourcing plan and constraints, in their final version.

**Audience:**

Yes, there would be a group of people in TMLR's audience be interested in knowing the findings of this paper.

**Claims And Evidence:**

Yes, the claims made in the submission are supported by clear evidence.